# *MRP3* is a sex determining gene in the diatom *Pseudo-nitzschia multistriata*

Monia T. Russo[1], Laura Vitale[1], Laura Entrambasaguas[1], Konstantinos Anestis [1], Neri Fattorini[1], Filomena Romano[1], Carmen Minucci[1], Pasquale De Luca[1], Elio Biffali[1], Wim Vyverman[2], Remo Sanges [1,3], Marina Montresor [1] & Maria I. Ferrante [1]

A broad diversity of sex-determining systems has evolved in eukaryotes. However, information on the mechanisms of sex determination for unicellular microalgae is limited, including for diatoms, key-players of ocean food webs. Here we report the identification of a mating type (MT) determining gene for the diatom *Pseudo-nitzschia multistriata*. By comparing the expression profile of the two MTs, we find five MT-biased genes, of which one, *MRP3*, is expressed exclusively in MT+ strains in a monoallelic manner. A short tandem repeat of specific length in the region upstream of *MRP3* is consistently present in MT+ and absent in MT− strains. *MRP3* overexpression in an MT− strain induces sex reversal: the transgenic MT − can mate with another MT− strain and displays altered regulation of the other MT-biased genes, indicating that they lie downstream. Our data show that a relatively simple genetic program is involved in defining the MT in *P. multistriata*.

[1] Stazione Zoologica Anton Dohrn of Naples, Villa Comunale, 80121 Naples, Italy. [2] Protistology and Aquatic Ecology, Department of Biology, Ghent University, 9000 Gent, Belgium. [3] Scuola Internazionale Superiore di Studi Avanzati (SISSA), via Bonomea 265, 34136 Trieste, Italy. These authors contributed equally: Monia T. Russo, Laura Vitale.  Correspondence and requests for materials should be addressed to M.I.F. (email: mariella.ferrante@szn.it)

Sexual reproduction is an ancient and almost universal feature in eukaryotes[1,2], and many types of sex determination systems have evolved in different eukaryotic groups[3,4]. The most studied systems are the XY or ZW sex chromosomes common in animals. However, in several multicellular organisms, sex can be determined by environmental cues, by multiple genes or by cytoplasmic elements[3]. Some organisms, such as mosses and brown algae, have haplo-diplobiontic life cycles and separate sexes are only found in the haploid phase of the life cycle[5,6]. In unicellular eukaryotes, mating can occur between gametes produced within the same strain (homothallic mating system) or between gametes produced by strains of complementary mating types (heterothallic mating system). The mating type is determined by a mating-type locus. Mating-type systems can be extremely sophisticated[7–9], illustrating the complexity and diversification of this fundamental process in biology.

To our knowledge, the genetic bases of sex determination are not known for diatoms, a species-rich group of photosynthetic protists that have a key role in the oceans and an extraordinary adaptive potential[10–12]. Diatoms have a life cycle characterized by a dominant diploid vegetative phase, in which they undergo mitotic divisions, and a short sexual phase, including meiosis, gametogenesis and syngamy, during which gametes are the only haploid stages. The diatom life cycle features a unique cell size reduction-restitution mechanism where the physical constraint imposed by the rigid silica cell wall determines a progressive reduction of the cell size of the population[13]. Sexual reproduction prevents extinction of the population because the zygote, which is not silicified, can expand, restoring the maximum large size. Large young cells are not competent for sexual reproduction and only cells below a sexual size threshold (SST) can undergo sex[13]. Diatoms are divided in two main groups, the ancestral centrics that are mainly planktonic, and the evolutionary younger pennates, mostly with a benthic lifestyle but including also a few ecologically important planktonic genera such as Pseudo-nitzschia and Fragilariopsis. Many centric diatoms are homothallic, and within the same clonal culture some cells differentiate into eggs and sperms that can self-fertilize, while most pennate diatoms are heterothallic and isogamous, or physiologically anisogamous[14], and sex occurs only when cells of distinct mating type (MT+ and MT−) below SST get in contact[13] (Supplementary Fig. 1). Evidence for the presence of sex pheromones in sexual reproduction was provided for the freshwater diatom Pseudostaurosira trainorii[15] and the marine benthic diatom Seminavis robusta[16,17]. Linkage mapping studies in the latter species identified an MT+ linkage group cosegregating with the MT, supporting a genetic basis for sex determination in diatoms[18]. Information on the molecular mechanisms governing diatom life cycles are however scarce, partly due to the fact that the laboratory strains of the two traditional model diatoms with the first sequenced genomes, Phaeodactylum tricornutum and Thalassiosira pseudonana, are putatively asexual[19,20].

In this work, we report the identification of a mating type determining gene in the marine planktonic diatom Pseudo-nitzschia multistriata. Pseudo-nitzschia is a cosmopolitan genus abundant in both coastal and open oceanic waters and comprising toxic species, responsible for human and animal poisoning events due to the production of the neurotoxin domoic acid (DA)[21–23]. P. multistriata is a DA producer, has a controllable and well described life cycle[24,25] and has been subject to extensive ecological and genetic investigations[26–28]. We recently sequenced its genome and elucidated the gene expression changes occurring during the early stages of sexual reproduction, highlighting an asymmetrical response of the two MTs to the sexual cues released in the process[29].

Here we used a transcriptomic approach coupled with genetic transformation to identify the MT determining gene, and attempt to reconstruct the genetic network underlying sex determination. The relatively simple model emerging from our study shows that MT+ is determined in strains < SST inheriting an active copy of the gene MRP3 that in turn controls expression of four other MT-related genes.

## Results

**Identification of mating type-related genes.** In P. multistriata, laboratory crosses yield a 50:50 ratio of MT+/MT−, suggesting genetic sex determination. In order to identify genes that were expressed uniquely in a given MT, we compared the expression profile of three MT+ and three MT− strains using RNA-seq. We found 35 differentially expressed genes (Supplementary Table 1), including 14 genes that were known to be induced during the early phases of P. multistriata sexual reproduction[29].

A subset of candidates was tested in qPCR on eight independent strains, different from those used to produce the RNA-seq data, and MT-restricted expression was confirmed for five genes, three specific to MT+ (0024820, 0122240, 0020770) and two to MT− (0085380, 0006960) (Table 1). We named these genes MRP1, MRP2 and MRP3 (Mating-type Related Plus) and MRM1 and MRM2 (Mating-type Related Minus), respectively. For the other candidates, expression levels were variable among strains but with no correlation to the MT (Supplementary Table 1).

None of the five MR genes has a clear function based on the annotation (Table 1). We analyzed the protein sequence of these five genes with software for the prediction of signal peptides[30,31] and found MRP1 to contain a putative signal peptide that targets proteins for translocation across the endoplasmic reticulum (ER) membrane (Supplementary Fig. 2), suggesting that the protein might travel through the ER and eventually might be targeted to the Golgi, to the vacuole or might be secreted[32]. Protein sequences of MRP2 and MRM2, predicted to be leucine-rich repeat (LRR) receptor-like protein kinases, were manually inspected and showed to possess a transmembrane region and a LRR domain but no recognizable protein kinase domain. The presence of a Heat Shock Factor (HSF)-type DNA-binding domain was confirmed in MRM1, while no domain could be found for MRP3.

Homology searches in public databases and in the MMETSP transcriptome dataset[33] revealed that the P. multistriata MT-biased genes were well conserved in species of the pennate diatom genera Pseudo-nitzschia and Fragilariopsis (Fig. 1a, b). For MRP1, a hit was detected also in Nitzschia punctata (a raphid diatom positioned in the clade above the red rectangle in Fig. 1a). For the LRR motif of MRP2 and MRM2 and for the HSF-type domain of MRM1, both common domains in diatom proteomes, hits were found in other diatom genera in the MMETSP database (Supplementary Data 1-3) however homology was limited to the domains and reciprocal blasts in the P. multistriata genome did not confirm a one-to-one relationship.

**Identification of the mating-type locus.** All MT-biased genes could be amplified from the genomic DNA of strains of both MTs. In some organisms, the genetic difference between the two sexes lies in the promoter of sex-determining genes[7], therefore we explored the intergenic regions upstream of the MR genes in the reference P. multistriata genome, obtained from an MT+ strain[29]. We noticed the presence of ambiguous bases in the MRM1 genetic region and of ambiguous bases and gaps in the MRP3 genetic region, indicating possible heterozygosity. Amplification and sequencing of a 728 bp region upstream of MRM1 from two MT+ and three MT− strains did not show differences linked to sex (Supplementary Fig. 3). For the MRP3 gene, an

**Table 1 The five mating type-related genes in *Pseudo-nitzschia multistriata***

| Gene name | Gene model | logFC in RNA-seq | logFC in qPCR ± variance | Description | Scaffold | Notes | URL[a] |
|---|---|---|---|---|---|---|---|
| *MRP1* | 0024820[b] | 3.06 | 9.8 ± 3 | — | 157 | Predicted signal peptide, unknown function, under positive selection | http://jbrowse.bioinfo.szn.it/pmultistriata/?loc=comp13283_c0_seq1 |
| *MRP2* | 0122240 | 4.33 | 3.7 ± 2.6 | LRR receptor-like serine/threonine-protein kinase GSO1 | 91 | Transmembrane LRR | http://jbrowse.bioinfo.szn.it/pmultistriata/?loc=PSNMU-V1.4_AUG-EV-PASAV3_0122240 |
| *MRP3* | 0020770 | 8.53 | 7.5 ± 5 | — | 147 | | http://jbrowse.bioinfo.szn.it/pmultistriata/?loc=PSNMU-V1.4_AUG-EV-PASAV3_0020770 |
| *MRM1*[c] | 0085380 0041130 | NA | −6.6 ± 0.8 | Heat shock factor protein 3 | 432 and 204 | | http://jbrowse.bioinfo.szn.it/pmultistriata/?loc=PSNMU-V1.4_AUG-EV-PASAV3_0085380 http://jbrowse.bioinfo.szn.it/pmultistriata/?loc=PSNMU-V1.4_AUG-EV-PASAV3_0041130 |
| *MRM2* | 0006960 | −8.17 | −9.7 ± 4.6 | Probable leucine-rich repeat receptor-like protein kinase At1g35710 | 11 | Transmembrane LRR | http://jbrowse.bioinfo.szn.it/pmultistriata/?loc=PSNMU-V1.4_AUG-EV-PASAV3_0006960 |

The gene name, gene ID, logFC (logarithmic fold-change) in RNA-seq and qPCR, the description of the predicted protein, the genomic scaffold and the link to the gene model in the genome browser are given
[a]Also available at http://gbrowse255.tgac.ac.uk/cgi-bin/gb2/gbrowse/maplesod_psnmu_v1_4_gbrowse255/, username: pnitzschia, password: 30DDFA0
[b]This gene model is incorrect, the corrected gene model can be found following the URL
[c]This gene (two identical gene models) was not present in the output of the differential expression analysis because of a filter on multimapping reads. It has been tested based on its expression profile during the early stages of *P. multistriata* sexual reproduction[29]

intronless gene located on scaffold 147 between positions 92,696 (predicted Transcription Start Site, TSS) and 93,904 (Fig. 2a), primers were designed to amplify a 1160 bp sequence upstream of the coding sequence (CDS), containing on the reference genome a stretch of 31 CTA triplets and 465 Ns (Fig. 2a). Amplification from MT+ and MT− strains showed different combinations of three bands of approximately 1110, 1050, and 1000 bp (Fig. 2b). The longest fragment was consistently present in all MT+ strains and absent in all MT- strains. We named the three alleles A (Alto, high), M (Medio, medium), and B (Basso, low), respectively. We confirmed that the A allele was a distinguishing feature of MT+ strains by amplifying this region in 86 strains, isolated from natural samples or obtained by laboratory crosses (Supplementary Table 2). In pedigrees, the A allele always segregated with the MT + phenotype (Supplementary Fig. 4). PCR products from 11 strains with different allelic combinations were cloned and sequenced, showing a variable number of CTA repetitions in the different alleles, the presence of GTA triplets in the M and B alleles, and GGA triplets in the B allele (Fig. 2e, Supplementary Fig. 5). Given that MT+ strains cannot be homozygous for the A allele (see Supplementary Fig. 4), the presence of a single 1110 bp band in some strains indicated the existence of an allele that had not been amplified. This hypothetical fourth allele was named N (Niente, nothing) (Fig. 2e). We sequenced additional fragments upstream of the triplet repeat region. In MT+ strains with a single 1110 bp band, when a second, overlapping fragment containing the triplet repeat region was amplified, again only one band could be seen with respect to strains with the A/M genotype in which two bands were seen (Fig. 2c). However, when the same forward primer was used in combination with a second reverse primer located upstream of the triplet region, all the strains showed two bands (Fig. 2d), demonstrating the existence of the fourth allele containing a region that we have not been able to characterize (Fig. 2e), possibly because of a chromosomal rearrangement preventing amplification. Alignment of the sequences of the four alleles showed the presence of several nucleotide variations (Supplementary Fig. 6).

The region around *MRP3*, for 20 kb upstream and 11 kb downstream, is rich in repetitive sequences and no coding genes can be found except for one gene (0020760) located 4 kb upstream of *MRP3* (Supplementary Fig. 7). 0020760 expression could be detected in both MTs (Supplementary Table 3). Synteny around the *MRP3* gene was explored in diatom species with a sequenced genome. The only diatom in which an *MRP3* homolog could be found is *Fragilariopsis cylindrus* (Fig. 1a). In this species, *MRP3* itself and the neighboring gene 0020760 lie isolated on two separate scaffolds and do not appear to retain synteny between them or with other neighboring genes, while synteny could be observed for genes upstream and downstream (Supplementary Fig. 8). Genes upstream of 0020760 and downstream of 0020770 also appear to retain synteny in the genome of *Pseudo-nitzschia multiseries*, in which 0020760 and 0020770 themselves could not be found (Supplementary Fig. 8).

An alignment of the MRP3 proteins found in public databases is shown in Supplementary Fig. 9.

**The A allele is predictive of the MT in cells above the SST**. In cells above the SST, the *MRP3* gene is not expressed (Supplementary Fig. 10), it must be therefore switched on when cells reach the SST. In order to look for any possible rearrangement in the putative sex locus between cells below and above the SST, we isolated 10 initial cells from new crosses and established 10 cultures of large (>SST) cells. The locus was amplified from the genomic DNA of these cells >SST and the results predicted five strains to be MT+, and five to be MT− (Supplementary Table 2). Only five of these cultures survived and, when these reached the SST, their predicted MT was confirmed by crossing them with strains of known MT (Supplementary Table 2). The locus was amplified from these strains below the SST and the pattern of alleles was confirmed, indicating that there are no genomic rearrangements in the putative sex locus when switching from sexually immature to sexually mature cells.

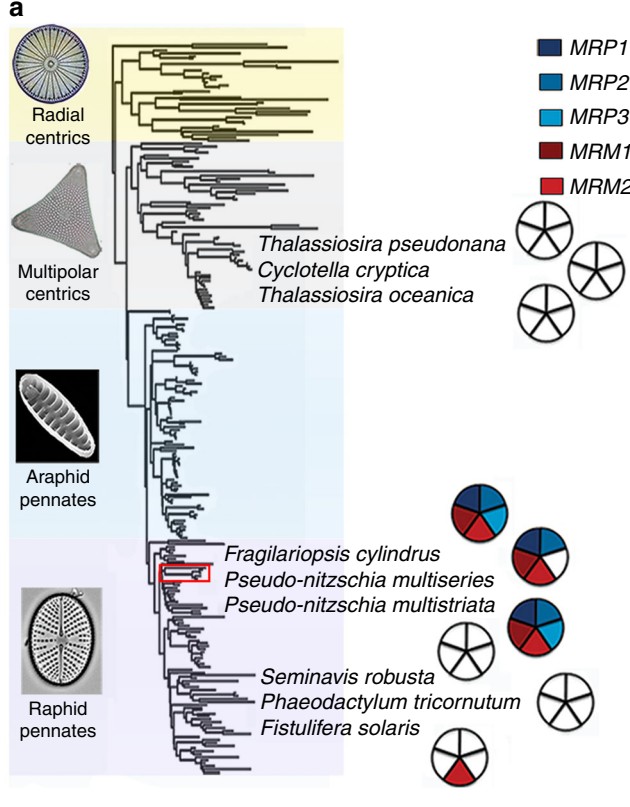

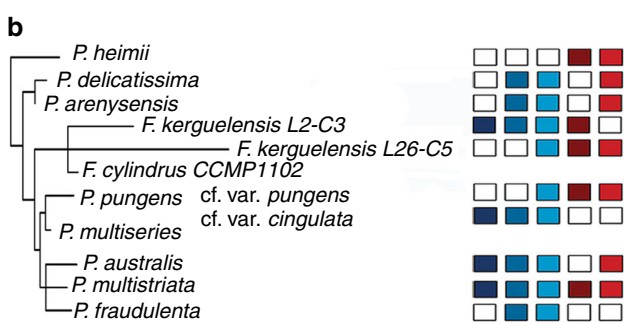

**Fig. 1** *P. multistriata* MT-biased genes conservation. **a** Conservation in diatom genomes. The diatom species with a sequenced genome are shown in a simplified diatom phylogenetic tree based on 18S (courtesy of W.H.C.F. Kooistra, modified from ref. [53]), color in the pie next to each species indicates the presence of a homolog, white indicates that no homology had been detected. Shades on the tree demarcate diatom lineages indicated on the left. The red rectangle indicates the area that is magnified in **b**. **b** Conservation in the transcriptomes of *Pseudo-nitzschia* and *Fragilariopsis* species

**Monoallelic expression of *MRP3* in MT+ strains**. We checked whether the expression of *MRP3* in MT+ could be monoallelic; in other words, we hypothesized that *MRP3* transcription occurs only on the A allele, present exclusively in MT+ strains. In the reference MT+ genome, *MRP3* showed seven SNPs. RNA-seq reads produced from the genome strain and from two other related MT+ strains contained only one of the possible two bases at each polymorphic position, indicating that only one of the two alleles was transcribed (Fig. 3a). We confirmed this result experimentally by sequencing a fragment of *MRP3* from the cDNA and genomic DNA of an additional MT+ strain (Fig. 3b). We reconstructed the two haplotypes from the sequence upstream of the triplet repeats to the CDS with three overlapping SNPs-containing PCR fragments, which were cloned and

sequenced. The sequence downstream of the repeats distinguishing the A allele corresponded to the sequence of the expressed *MRP3* allele (Fig. 3c).

**Overexpression of *MRP3* in MT− strains and sex reversal**. The CDS of *MRP3*, amplified by RT-PCR from the MT+ strain LV149, was cloned downstream of a constitutive *P. multistriata* promoter[34] and transformed in the MT− strains LV136 and F4B2. Integration of the transgene in two resistant strains, LV136T3 and F4B2T7, was verified by PCR on genomic DNA (Fig. 4a and Supplementary Fig. 11) and exogenous expression by RT-PCR (Fig. 4a). As all MT- strains, the nontransformed LV136 did not express *MRP3*, while the transformed LV136T3 strain did express it. The SNP profile of the RT-PCR product revealed that the expressed *MRP3* transcript derived from the transgene and not from the endogenous gene (Supplementary Fig. 12A).

The LV136T3 and F4B2T7 strains and a control strain transformed only with the resistance gene were tested in crosses. Differently form the nontransformed strains LV136 and F4B2, and from the control strain, LV136T3 and F4B2T7 produced sexual stages when crossed with MT− strains but not with MT+ strains, indicating sex reversal (Fig. 4b and Supplementary Fig. 11B). F1 strains were isolated from LV136T3 x MT− crosses and, as expected, all displayed only M and B alleles (Fig. 4c). Eight of these F1 cultures survived until they reached the SST, at this stage they were tested in crosses and all behaved as MT−. Three out of these eight displayed inheritance of the *MRP3* transgene which for unknown reasons was however not expressed (Supplementary Fig. 13), explaining lack of sex reversal.

More detailed analyses were conducted on LV136T3. Sex reversal in this strain was accompanied by an inversion in the pattern of expression of the other MT-biased genes: *MRP1* and *MRP2* were upregulated, while *MRM2* was downregulated (Fig. 4d). qPCR quantification showed a reduction of 4.3-fold and 5.4-fold for *MRM1* and *MRM2* expression, respectively, in LV136T3 compared to nontransformed LV136, values comparable to those measured for a wild-type MT+ strain (Fig. 4e). These data indicate that *MRP3* is upstream of the other MR genes.

Finally, in order to verify whether *MRP3* overexpression had affected other genes, we compared the gene expression profiles of nontransformed vs transformed LV136 with RNA-seq. Only seven genes were found to be significantly differentially expressed, and these included *MRP3* and the other four MR genes (Supplementary Table 4). Of the remaining two genes, both weakly expressed, one, named *MRX1*, has a putative DNA repair domain and the other, named *MRX2*, is an unknown protein. The role of these genes remains to be evaluated.

Inspection of the RNA-seq reads produced for the transformed LV136T3 further confirmed that the expressed *MRP3* is the exogenous gene, based on the SNPs content and on the absence of reads for the 5′ and 3′ UTRs (Supplementary Fig. 12B), and also allowed to verify that the global SNPs profile of LV136T3 was identical to the SNPs profile of LV136.

## Discussion

Our study showed that in the diploid diatom *Pseudo-nitzschia multistriata* the mating type identity is defined by the monoallelic expression in the MT+ of the gene *MRP3*, with unknown function, which regulates the expression of two MT+ and two MT− biased genes. Evidence was provided by the reconstruction of the allelic variants of *MRP3*, by the demonstration that only the A allele is expressed in MT+, and by the production of transgenic strains with *MRP3* overexpression.

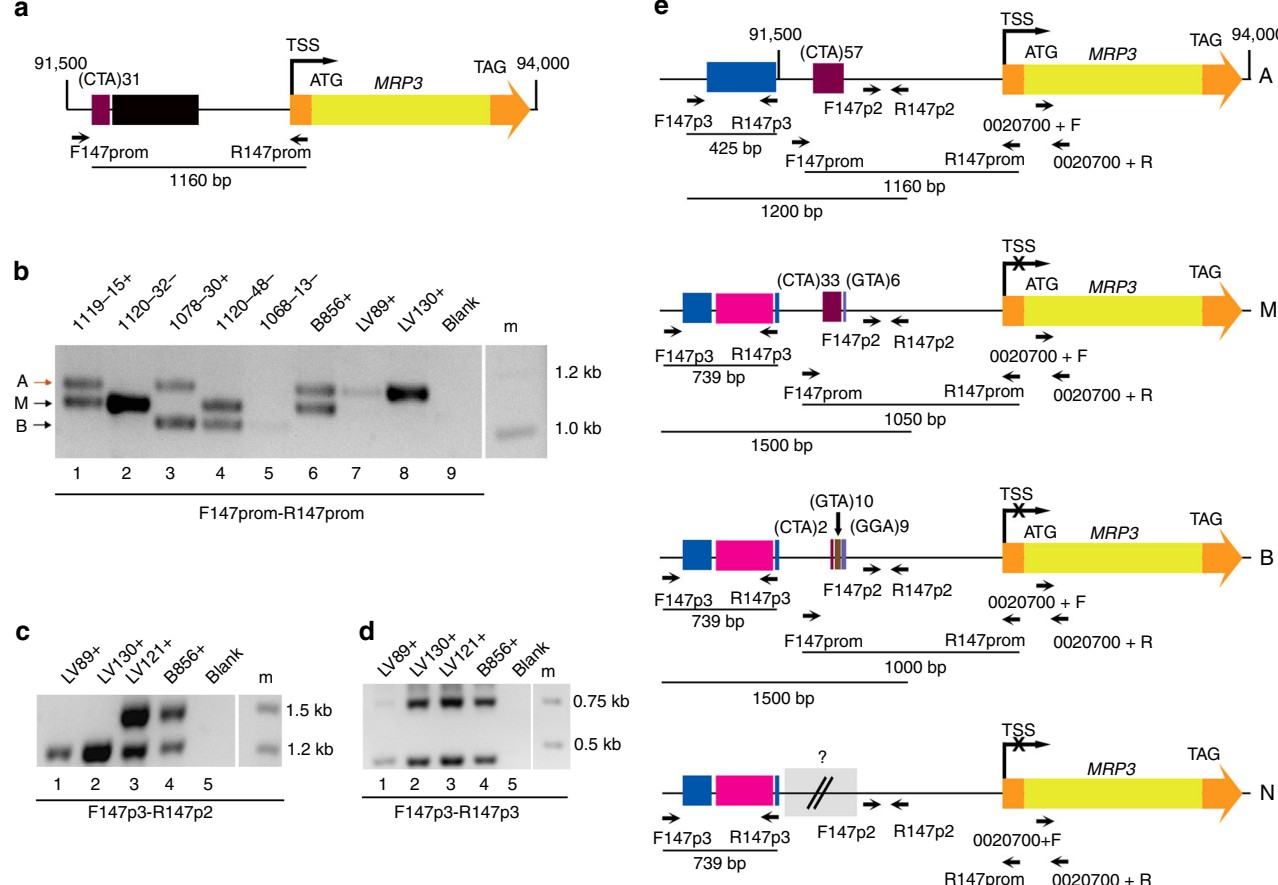

**Fig. 2** *MRP3* alleles. **a** Schematic representation of the genomic region between 91,500 and 94,000 bp on scaffold 147. The intronless gene *MRP3* is represented as a thick arrow with the CDS in yellow and the UTR regions in orange, the transcription start site (TSS), the start codon (ATG) and the stop codon (TAG) are indicated. Primers are represented as black arrows. The purple box represents CTA repeats and the black box a stretch of Ns. **b** Electrophoretic gel with amplicons obtained from five MT+ and three MT− strains. The high band (A, orange arrow) is present in MT+ only, m stands for marker. **c** Electrophoretic gel with amplicons obtained using primers F147p3-R147p2 from four MT+ strains which, with the PCR shown in **b**, had only the A band (LV89 and LV130), or the A and the M band (LV121 and B856). **d** Electrophoretic gel with amplicons obtained from the same four MT+ strains as in **c**, using a different reverse primer. **e** Schematic representation of the A, M, B, and N alleles. The purple, brown and violet boxes represent CTA repeats, GTA repeats and GGA repeats respectively, the number of repetitions is indicated for each triplet. The pink box represents a repetitive sequence, the blue box remnants of a transposase. The sequence in the gray box could not be amplified

There are limited examples of the mechanisms of sex determination in algae. In green algae, the well-studied unicellular *Chlamydomonas reinhardtii* and the multicellular *Volvox carteri* have both a haploid system of determination with characterized molecules and a certain degree of conservation[35]. In *C. reinhardtii* the sex-determining gene is the *MID* transcription factor expressed by the haploid MT− gametes. The MT loci have been characterized and contain inversions and translocations that contribute to suppress recombination. Green algae are however evolutionarily distant from diatoms. The closest relative to diatoms with known information on sex determination is the stramenopile brown macroalga *Ectocarpus siliculosus*[36]. This species features two alternating life history phases, a diploid sporophyte and haploid male and female gametophytes. The male and female gametophytes morphological differences are correlated to the existence of a few dozens of sex-related genes[37]. Sex is determined during the haploid phase with a UV system, with a male V chromosome and a female U chromosome. The *E. siliculosus* candidate sex-determining gene contains a male-specific high-mobility group (HMG) domain[6], that is often found in proteins known to be involved in sex determination systems[38,39]. As expected for sex-determining regions, also in the *E. siliculosus* sex locus the presence of transposable elements and of fewer coding

genes compared with other regions of the genome are compatible with the loss of recombination. In other brown algae sexual dimorphism is expressed during the diploid phase[40] and, although a putative sex-determining region has been identified in *Laminaria*[41], no sex determination system has been described yet for dioecious brown algae. As frequently observed also in animals, plants, and fungi[3,4], known algal systems display heterogeneous solutions with limited conservation. Diatoms are unicellular members of the stramenopile lineage and *MRP3* is conserved in other *Pseudo-nitzschia* species and in the closely related *Fragilariopsis* (Fig. 1), although it remains to be determined whether *MRP3* acts as MT determinant also in these species.

Homology searches failed to detect a homolog of *MRP3* in centric and other pennate diatoms, including *Seminavis robusta*. This can be due to a true absence, to rapid evolution that might hamper its recognition based on sequence homology, or to technical issues with the genome assembly, such as a difficulty to assemble the genomic regions in which it is contained, due to the high number of repetitive sequences typical of sex-determining regions.

A true absence and/or lack of conservation in the mechanism would imply that different systems to determine the MT evolved independently within diatom lineages, a finding that would be in

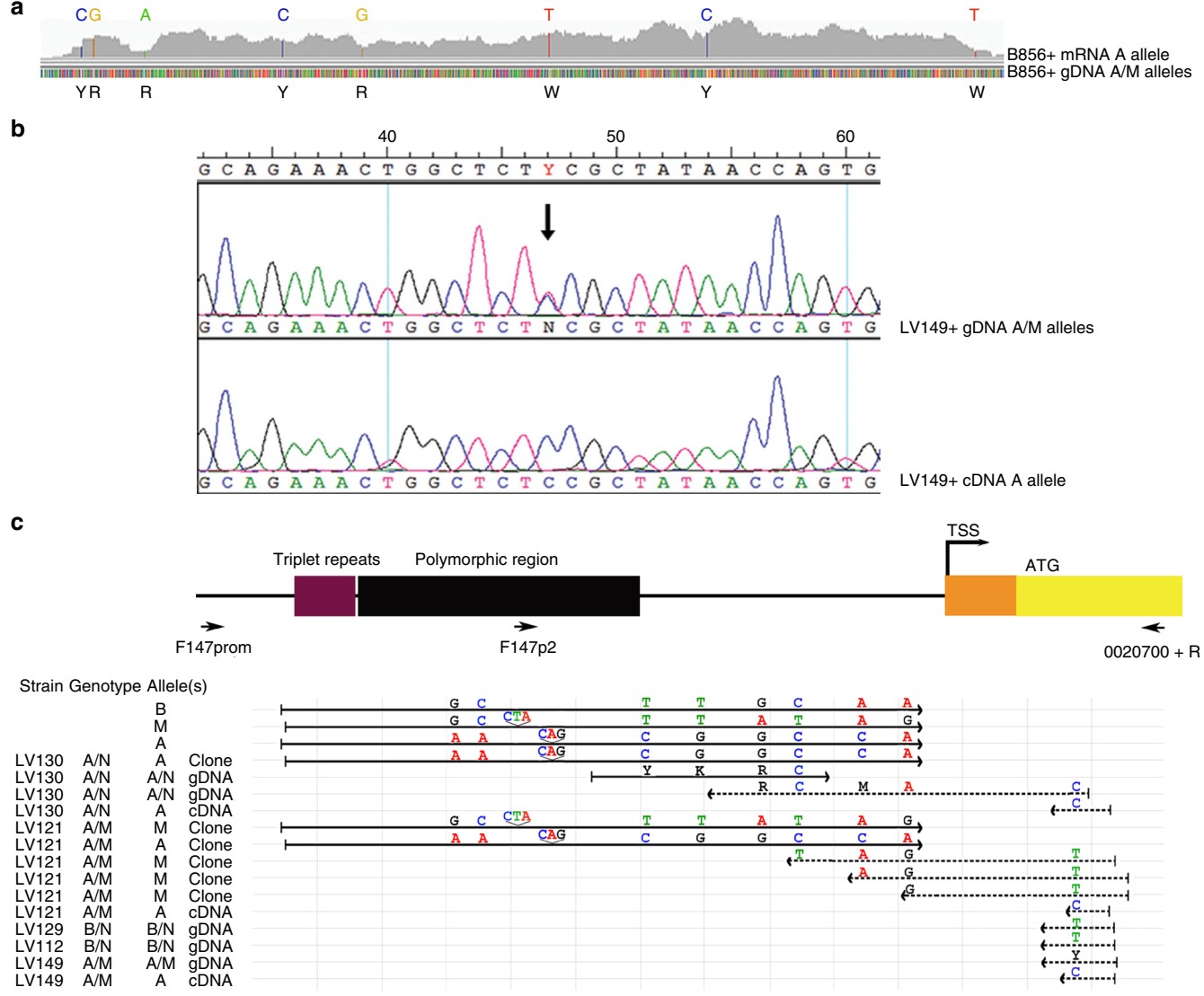

**Fig. 3** Monoallelic expression of *MRP3* is linked to the A allele. **a** IGV (Integrative Genomics Viewer) visualization of the *MRP3* gene on scaffold 147. Colored vertical lines at the bottom represent the DNA nucleotides in the genome, with SNPs indicated below. The gray peaks above correspond to the RNA-seq reads from strain B856 (MT+, A/M genotype). Colored bars are in correspondence of nucleotides that differ between the reference DNA sequence and the transcript sequence. **b** The top electropherogram shows the sequence of an *MRP3* fragment amplified from the gDNA of strain LV149 (MT+, A/M genotype), the bottom track is from the cDNA of the same strain. A black arrow indicates the double peak in the gDNA. **c** Schematic representation of the alignment of Sanger sequences from PCR fragments obtained amplifying either gDNA or cDNA, or from different TOPO vectors in which alleles were cloned. The strain name, its genotype and the type of sequence are shown on the left. The consensus sequences of the A, M, and B alleles were included in the alignment for reference. A schematic representation of the genomic region under study is shown above the alignment (see Fig. 2a), with the primers used for sequencing indicated. Polymorphic nucleotides or indels are indicated at their approximate position. Note that the *MRP3* transcript shows a C, which is on the A allele, while T, the alternative nucleotide in that position, is on the M allele

line with the plasticity of sex-determining mechanisms recorded also amongst phylogenetically related species[3,4].

Only a specific allele, named A, is linked to *MRP3* expression, and we identified at least three other alleles with structural differences, linked to a lack of *MRP3* expression (Fig. 2). In addition to the specific microsatellites in the proximity of the gene, the 32 kb genomic region around it is gene poor and enriched in repetitive sequences.

Several reports indicate that microsatellites can have a functional role acting in processes linked to gene expression regulation[42]. Microsatellite regions can be inherently unstable, expanding or contracting with mitotic DNA replication, but again whether this plays a role in the regulation of the putative sex locus remains to be determined. Microsatellites can be bound by transcription factors, can modify the distance between promoter elements or enhancers and promoters, can induce changes in the DNA conformation affecting accessibility, and even cause epigenetic disruption[42]. The exact role of microsatellites in the *MRP3* alleles is not clear, but one possible hypothesis is that they take part in the control of *MRP3* expression. The A allelic configuration, with the longest stretches of CTA, could be linked to an open status of the chromatin allowing access of the transcriptional apparatus and expression of the gene, while the other configurations could be heterochromatic and inaccessible to the transcriptional apparatus. Heterochromatinization, which we hypothesize for the M and B alleles, is one of the mechanisms that allows the maintenance of suppression of recombination[43].

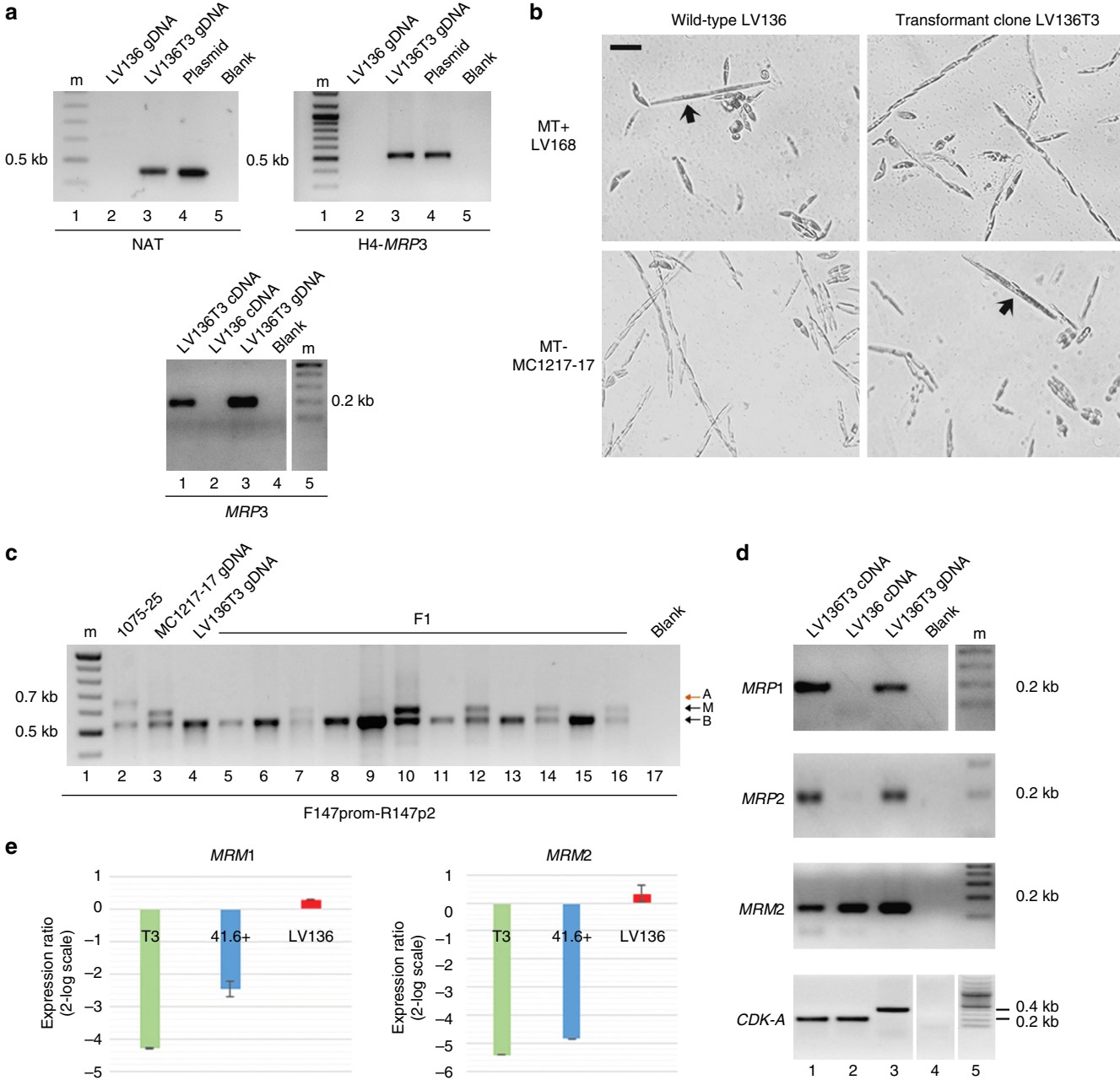

**Fig. 4** Sex reversal in the MT- LV136 transformed strain overexpressing *MRP3*. **a** On the top, amplification of a fragment of the nourseothricin resistance gene (primers NATfor/NATrev, left) and a fragment of the transgene from the H4 promoter to the *MRP3* CDS (primers H4up1-0020770+R, right) from the genomic DNA of strain LV136T3 is shown; on the bottom, amplification of a fragment of the *MRP3* transcript (primers 0020770+F/0020770+R for *MRP3*) from the cDNA of strain LV136T3 is shown. m indicates the 100 bp DNA ladder. **b** Images of crosses of wild-type LV136 (MT−) and transformant LV136T3 with LV168 (MT+) and MC1217-17 (MT−). Wild-type cells mated with LV168 producing initial cells (the longest cells indicated by arrows), while the transformant cells did not. Opposite results were observed in crosses with MC1217-17. Scale bar = 20 μm. **c** PCR analysis for the variable region of the *MRP3* alleles on gDNA of F1 siblings obtained from the cross between MC1217-17 (MT−) and LV136T3. 1075-25 is a control MT+ strain. **d**, **e** The LV136T3 strain displays amplification of MT+ specific genes (top two gels) and less intense signals for MT− specific genes with respect to the nontransformed LV136 strain (bottom gel and qPCR graphs). Amplification of an intron-containing *CDK-A* fragment is shown to demonstrate that the cDNAs are free from genomic contamination. In the qPCR graphs, expression levels of LV136T3, of the MT+ control strain 41.6 and of LV136 are relative to expression levels of an MT− control (strain 41.4) set as 0 (see Methods). Error bars represent the standard deviation of three biological replicates

The model emerging from our data suggests that when *P. multistriata* cells reach the SST in the absence of the A allele, *MRM1* and *MRM2* are expressed specifying the MT−. Inheritance of the A allele on the other hand leads to *MRP3* expression, which in turn determines MT+ biased expression of *MRP1* and *MRP2* and downregulation of *MRM1* and *MRM2*, specifying the MT+.

MRM1, lowly expressed and with a HSF DNA-binding domain, could regulate transcription or the chromatin status and specify the MT− as a default state. MRP1 contains a signal

peptide for organelle targeting or secretion and, in our previous studies, we had found it among the most highly induced genes when the MT+ perceives signals from the MT− during the early phases of sexual reproduction[29]. We speculate that the product of this gene is a secreted molecule involved in sexualization or attraction of the MT− cells. While MRP3 governs the network to specify the MT, MRP1 would therefore be required for MT+ cells to be sexually competent. It is also worth noting that this gene has been found to be under positive selection[29]. The MRP2 and MRM2 proteins contain predicted transmembrane domains and LRR domains, and we hypothesize that they could be, respectively, the MT+ and MT− pheromone receptors or adhesion molecules.

Together with our previous findings on the diatom meiotic toolkit[19], this new set of genes can be added to the repertoire of markers that will enable the exploration of the process of sexual reproduction at sea, by browsing metatranscriptomics data deriving from many sources, including the recently released Tara Oceans expedition dataset[44]. Moreover, the results of this study will help to discern the mechanisms that rule the demographic structure of natural diatom populations. *P. multistriata* blooms in the Gulf of Naples show high genotypic diversity. While ~50 MT +:50 MT− ratios are normally observed, in the course of 2015, a clonal demographic expansion dominated by MT+ strains was detected[28]. Whether this is common in diatom blooms and if and how this relates to toxin levels in harmful algal blooms[21] remains to be determined. Knowledge and control of life cycle dynamics will help to understand unexplained observations and will be particularly relevant for toxic species responsible for harmful algal blooms.

Research on diatoms is intensifying as they are recognized as the most diverse and plastic components of phytoplankton, and also as promising organisms for biotechnological applications. So far, the lack of a sexual phase in the most studied species has hampered the full development of diatoms as genetic model systems. Our data provide a baseline for future studies of processes including diatom gene expression regulation, molecular evolution, diversification of sex determination mechanisms, population dynamics and speciation mechanisms.

## Methods

**Strains**. *Pseudo-nitzschia multistriata* cells were grown in f/2 medium[45] at 18 °C, 50 μmol photons m$^{-2}$ s$^{-1}$ and a 12L:12D h photoperiod. Strains were either isolated from the LTER station MareChiara in the Gulf of Naples[26,28] or obtained from laboratory crosses[29] (Supplementary Table 2). Strains B856, B857, Sy373 and Sy379, were used to produce a de novo transcriptome.

**Production of an F1 population**. A full-siblings family was produced by crossing strains B854 MT+ and MVR1041.4 MT−. The two strains were mixed at a final cell concentration of 5000 cells ml$^{-1}$ for each strain. The co-culture was inoculated in 2-ml wells of culture plates and incubated at 20 °C and 130 μmol photons m$^{-2}$ s$^{-1}$, using as light source cool white fluorescent tubes TLD 36W/950 (Philips, Amsterdam, The Netherlands). Large F1 initial cells were produced after about 2 days and two initial cells for each well were manually isolated with a micropipette. A total of 150 initial cells were individually transferred into 24-well culture plates with 2 ml f/2 medium. Once the cultures reached sufficient density, they were transferred to tissue culture flasks containing 20 ml of f/2 medium. After almost 4 months of weekly transfers, the F1 cultures reached the sexualization cell size threshold (cell length < 60 μm). The mating type was determined for 110 survived F1 strains by crossing each individual strain with two MT+ and two MT− reference strains: MT− SH20 and MVR171.1, and MT+ MVR171.8 and B856. The mating type of 93 F1 strains was determined, yielding a population of 41 MT+ and 52 MT− strains.

**Sequencing data and bioinformatic analyses**. For the gene expression analysis, we used RNA-seq data from nine libraries, from a total of six strains (MT+ Sy373, B856, B938, MT− Sy379, B857, B939) (Supplementary Table 5). A quality check was performed on the raw sequencing data before proceeding to further analyses, removing low quality portions. The minimum accepted length was 35 bp and the quality score 20.

The high quality reads were aligned against the *P. multistriata* reference genome sequence (version 1.4) with STAR aligner[46] (version 2.5.0c). On average, 90% of reads could be uniquely mapped on the genome in all samples except for CIIO1, CIIO2, and CIIP2, where the percentage dropped from 50% to 70%. FeatureCounts (version 1.4.6-p5) was used in combination with the *P. multistriata* annotation to calculate gene expression values as raw read counts. To apply normalization to the raw read counts we used the Trimmed Mean of M-values (TMM) normalization. RPKM/FPKM values were also calculated. For all the statistical analyses we used R with the packages HTSFilter and edgeR[47].

For quality control, as first step we removed non-expressed genes and the ones showing too much variability within the biological replicates. For this purpose, we chose the HTSFilter package, which implements a filtering procedure for replicated transcriptome sequencing data based on a Jaccard similarity index. These steps removed 134 transcripts. Transcripts passing HTSFilter were used for differential expression analysis between MT+ and MT− using a generalized linear model (GLM). Genes were considered significantly differentially expressed if the false discovery rate (FDR) of the statistical test is less than 0.05. No filter to the fold-change was used, however all the significantly differentially expressed genes showed a log2 fold-change of at least 2.

**Sample collection and RNA extraction**. Cell growth was followed by estimating cell concentration using a Malassez counting chamber. Cells were collected in exponential phase onto 1.2 μm pore-size membrane filters (RAWP04700 Millipore), and extracted with Trizol™ (Invitrogen) following manufacturer's instructions, a DNase I (Qiagen) treatment was applied to remove the gDNA contamination, and RNA was further purified using RNeasy Plant Mini Kit (Qiagen). RNA quantification was done using Qubit® 2.0 Fluorometer (Life Technologies) and integrity was assessed using Bioanalyzer (2100 Bioanalyzer Instruments, Agilent Technologies).

**RT-PCR and qPCR**. From 1 to 0.25 μg of total RNA was reverse-transcribed using the QuantiTect® Reverse Transcription Kit (Qiagen, Venlo, Limburgo, Nederlands). To assess cDNA quality and absence of gDNA contamination, RT-PCRs were run with control genes *TUB A* and *CDK-A*[48]. The uncropped versions of the gels can be found in Supplementary Note 1.

Real time PCR amplification was performed with cDNA diluted 1:5, in a 10 μl reaction containing each primer at a final concentration of 1 μM and Fast SYBR Green Master mix with ROX (Applied Biosystems). Reaction were run in a ViiA™ 7 Real-Time PCR System (Applied Biosystems by Life Technologies, Carlsbad, CA, USA). Cycling parameters were: 95 °C for 20 s, 40 cycles at 95 °C for 1 s, 60 °C for 20 s, 95 °C for 15 s, 60 °C 1 min, and a gradient from 60 °C to 95 °C for 15 min. Raw results were processed using the ViiA™ 7 Software and exported into Microsoft Excel for further analyses. Biological triplicates were used for all samples and each biological replicate was done in technical triplicate.

To calculate primer amplification efficiency we made a serial 10-fold dilution and used the Standard Curve method of the ViiA™ 7 Real-Time PCR System. The reference genes used in the qPCR were *TUB A*, *TUB B* and *CDK-A*[48]. Fold-changes were obtained with the Relative Expression Software Tool-Multiple Condition Solver (REST-MCS)[49].

**Conservation and sequence analyses**. The predicted proteins of the MT-biased genes were used in tblastn searches against nr, against the Marine Microbial Eukaryote Transcriptome Sequencing Project dataset (https://www.imicrobe.us/#/search/mmetsp)[33] and against the diatom genomes available. The *Thalassiosira pseudonana* CCMP 1335, *Phaeodactylum tricornutum* CCAP1055/1, *Fragilariopsis cylindrus* CCMP 1102 and *Pseudo-nitzschia multiseries* CLN-47 genomes were found at the JGI website (http://genome.jgi.doe.gov/), for *Thalassiosira oceanica* and *Fistulifera solaris* gene models are in NCBI, whereas for *Cyclotella cryptica* a local blast was used downloading the assembly from http://genomes.mcdb.ucla.edu/Cyclotella/download.html. The proteins were also blasted against a *Seminavis robusta* de novo genome sequence. We also searched in the genome of the stramenopile macroalga *Ectocarpus siliculosus* (http://bioinformatics.psb.ugent.be/orcae/overview/Ectsi). Retrieved protein sequences were classified as homologs based on the significance (e-value < 1e10$^{-3}$) and on the percentage of sequence identity (>30%). The protein products of the orthologous genes were retrieved, manually checked and validated with a reciprocal tblastn on the *P. multistriata* genome.

Convincing hits were found for *Pseudo-nitzschia* and *Fragilariopsis* species, while for other diatoms species, when hits were found, homology was limited to the conserved LRR domain of MRP2 and MRM2 and to the HSF domain of MRM1 (Supplementary Data 1-3). For Fig. 1, when the reciprocal tblastn in the *P. multistriata* genome did not confirm a one-to-one correspondence, the hit was not considered. For *P. arenysensis*, in a transcriptome recently obtained in our laboratory, we detected the presence of *MRP2* and *MRP3* (Supplementary Fig. 14) which had not been detected in the MMETSP *P. arenysensis* transcriptome. Note that no information on the MT of the MMETSP sequenced strains is available and that for some species more than one strain has been sequenced, making the correlation of the presence/absence of the MR-biased genes with the putative MT unreliable.

To look for signal peptides, we used two software programs: (i) SignalP 4.0, which can predict the presence of a secretory signal peptide, a ubiquitous protein sorting signal required for protein translocation across the endoplasmic reticulum (ER) membrane[31]; (ii) AsaFind, a prediction tool that identifies nuclear-encoded plastid proteins in algae with secondary chloroplasts of the red lineage, based on the presence of the conserved 'ASAFAP' motif and transit peptide[30].

For the synteny analysis, protein sequences of all genes present on *P. multistriata* scaffold 147 were used as queries in tblastn searches in the genomes of *F. cylindrus* and *P. multiseries*. Scaffolds with hits were retrieved and used for the graphical representation shown in Supplementary Fig. 8 obtained with the graphical program SympleSynteny[50].

Sequence alignments were made with Clustal Omega[51].

**DNA extraction and PCR analyses.** Genomic DNA was isolated using cetyl-trimethylammonium bromide (CTAB) or phenol. In the first protocol, 500 μl CTAB and 12 μl β-mercaptoethanol were added to the cell pellet, then 4 μl RNase were added and the reaction was incubated at 65 °C for 45 min, the sample was mixed by vortexing. After addition of 500 μl SEVAG (chloroform: isoamyl alcohol 24:1) the sample was centrifuged at 20,000×*g* for 15 min at 4 °C. This step was repeated, 1 volume of ice-cold isopropanol was added and the reaction was incubated at −20 °C overnight. The sample was centrifuged at 20,000×*g* for 30 min at 4 °C and the pellet was washed with 400 μl 75% ethanol. After a further centrifugation at 20,000×*g* for 15 min at 4 °C, the pellet was air-dried and finally resuspended in 20 μl of distilled water and incubated overnight at 4 °C before checking quality and quantity. In the phenol method, cells (300 ml of culture with cell density 80,000 cells ml$^{-1}$) were harvested by filtration onto 1.2 μm pore-size filter (RAWP04700 Millipore), rinsed with f/2 medium, transferred to tubes and centrifuged at 4000×*g* for 5 min at 4 °C. 400 mg of glass beads 0.2–0.3 mm diameter (G1277, Sigma-Aldrich), 500 μl of Tris-EDTA buffer and 500 μl phenol were added to the cell pellets and the mixtures were vortexed 30 Hz 3 times for 85 s. After centrifugation at 10,000×*g* for 5 min at 4 °C, aqueous phases were recovered, 500 μl of phenol: chloroform: isoamyl alcohol 25:24:1 v/v mixture was added and the tubes were mixed by inversion. After another step of centrifugation, 50 μg of RNase A were added and the tubes were incubated at 37 °C for 30 min. Another step of phenol: chloroform: isoamyl alcohol extraction was followed by overnight precipitation at −20 °C in the presence of 3 M sodium acetate (pH ± 5), ethanol 96% and glycogen. Samples were then centrifuged at 18,000×*g* for 30 min at 4 °C. Pellets were washed in the presence of ethanol 70% and centrifuged at 18,000×*g* for 5 min at 4 °C. The pellets were air-dried and resuspended in sterile MilliQ water.

PCRs were performed by using Q5® High-Fidelity DNA Polymerase (New England Biolabs) according to manufacturer's instructions. Primer sequences are reported in Supplementary Table 6. The uncropped versions of the gels can be found in Supplementary Note 1.

**Subcloning and Sanger sequencing.** Amplicons were subcloned using TOPO TA Cloning® (Invitrogen™) when two (or more) products were obtained or directly sequenced in the case of a single product. Sanger sequences were performed by using the corresponding primers pair used to generate the PCR product.

**Plasmid construction.** To prepare the PmH4pMRP3At overexpression construct, FMrp3Eco, containing the *Eco*RI restriction site, and RMrp3Sma, containing the *Sma*I restriction site, were used to amplify the *MRP3* coding sequence from the cDNA of strain LV149. The PCR product and the construct PmH4pGUSAt[34] were digested with *Sma*I at 25 °C and then with EcoRI-HF at 37 °C in the Cutsmart buffer (New England Biolabs).

The digested construct was dephosphorylated using Alkaline Phosphatase, Calf Intestinal (CIP) (New England Biolabs) and both the construct and the insert were purified with QIAquick PCR Purification Kit and ligated by using Quick Ligation™ Kit.

To prepare the PmH4NatAt, containing the antibiotic resistance expressing cassette, FPIH4 and RPIH4 primers were used on PmH4BleAt[34] to amplify the plasmid backbone containing the PmH4p and the PtAt, whereas the primers FPIIH4 and RPIIH4 were used on pBNat to amplify the *Nat* coding sequence. To design the primers and perform the PCR, the NEBuilder tool (http://nebuilder.neb.com/) and the Gibson Assembly kit were used as suggested by manufacturer's instructions. Primers sequences are reported in Supplementary Table 6.

**Genetic transformation.** Genetic transformation was performed as described in ref. [34] and transformant clones were selected using 300 μg ml$^{-1}$ nourseothricin (clonNAT WERNER BioAgent GmbH, Jena, Germany) or with 1 μg ml$^{-1}$ zeocin (Invivogen).

**Gene expression analysis of the LV136T3 transformant strain.** RNA was extracted from LV136T3 and LV136 grown in triplicate. RNA quality and quantity were evaluated with the Nanodrop and with the Plant RNA Nano® assay in an Agilent Bioanalyzer®. Two replicas were used for RNA-seq analysis. Sequencing was performed using the Ion P1 sequencing Kit v2 (Thermo Fisher Scientific) on an Ion Proton™ sequencer, following Manufacturer's instructions. Samples were run

on two Ion P1 Chip Kit v3 chips (Thermo Fisher Scientific). All the libraries were sequenced in both chips to prevent the influence of batch effects.

Quality control of sequencing reads was initially done using FastQC v0.11.5 (http://www.bioinformatics.babraham.ac.uk/projects/fastqc). Raw reads were trimmed and quality filtered using Fastx_toolkit v0.0.14 (http://hannonlab.cshl.edu/fastx_toolkit/). Low quality extreme nucleotides were trimmed from the ends and only reads with a quality score ≥ 20 and a minimum length of 50 bp were retained. FastQC was performed again to verify the integrity of the remaining raw sequence reads. Between 69% and 79% of initial raw reads were used for further analysis.

Filtered reads were mapped against the *P. multistriata* genome v1.4 using STAR aligner v2.5.3a[46] (parameters: --outFilterMismatchNoverLmax 0.05 --outFilterMultimapNmax 7 --outFilterMatchNminOverLread 0.66 --outSJfilterCountUniqueMin 5 5 5 5 --outFilterType BySJout BySJout --outFilterIntronMotifs RemoveNoncanonical –seedSearchStartLmax 25 --seedMultimapNmax 1000). On average, 76.4% of the reads mapped uniquely to the genome. The number of reads for each gene was counted by multiBamCov v2.17.0 of BEDTools[52]. Differential gene expression analysis between control and the transformed strain was conducted using the edgeR package v3.16.5[47] in R. Raw counts were filtered and only genes with CPM values above 1 in at least two libraries were included in the analysis. Gene counts were then normalized with the weighted trimmed mean of M values (TMM) method in edgeR. Genes were considered significantly differentially expressed when their expression values had a fold-change greater than ±1 and a FDR-corrected *p* value of ≤0.05 (Benjamini–Hochberg step-up correction).

**Reporting summary.** Further information on research design is available in the Nature Research Reporting Summary linked to this article.

## Data availability

The data generated or analyzed during this study are included in this published article and its Supplementary Information files. Sequencing data are deposited in the Sequence Read Archive database (http://www.ncbi.nlm.nih.gov/Traces/sra/) under the accession number MTAB-7121. *P. multistriata* genome data are available at http://bioinfo.szn.it/pmultistriata/. For accession numbers and links to other data obtained from publically available sources, refer to the Methods section and to Supplementary Table 5.

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

## Acknowledgements

The authors thank A. Storlazzi, M. Chiurazzi, M. Ribera D'Alcalà, C. Bowler and C. Conicella for valuable comments, W.H.C.F. Kooistra for the diatom tree in Fig. 1. F. Manfellotto, E. Mauriello, R. Pannone, and A. Manfredonia for support, R. Aiese Cigliano and W. Sanseverino (www.sequentiabiotech.com) and the SZN Bioinforma service for bioinformatics support. L.V. was supported by an OU SZN PhD fellowship. The work was partially supported by EMBRIC G.A. 654008. The work conducted by the U.S. Department of Energy Joint Genome Institute is supported by the Office of Science of the U.S. Department of Energy under Contract No. DE-AC02-05CH11231.

## Author contributions

M.M., W.V., M.I.F., R.S., M.T.R., and L.V. designed the study. M.I.F., L.V., M.T.R., K.A., N.F., and F.R. performed the experiments. C.M. provided technical support. P.D.L. and E.B. sequenced the transformant. L.E. and R.S. performed the bioinformatics analyses. M.I.F., M.M., L.V., M.T.R., R.S., and W.V. analyzed the data. M.I.F., M.M., and M.T.R. wrote the paper. All authors were involved in reading and correcting the manuscript.

## Additional information

**Competing interests:** The authors declare no competing interests.

