## [Peer Review File · Nature Communications]

Reviewers' Comments:

Reviewer #1:

Remarks to the Author:

The manuscript by Russo et al. describes their discovery of a mating type locus for an important member of the marine diatoms – *Pseudo-nitzschia multistriata*. Diatoms have an unusual sexual cycle: responsiveness to an external induction trigger is linked to a reduction in cell size resulting from sequential mitotic divisions. The assumed paradigm is that cell lines will die off if sexual reproduction does not occur with a resultant cell enlargement. This is a fascinating example of the diversity of unicellular eukaryotes, with important evolutionary ramifications. The data underlying the manuscript is very solid. As described below, however, the description of some experimental results could be better clarified and the overall results placed into a larger context.

1) Heterothallic diatoms, which are the subject of this manuscript, have two distinct mating types that display Mendelian inheritance patterns. A second large group of diatoms have a homothallic life cycle where any individual cell can become either male or female gametes. The authors should place their work within this larger context. What does it mean that only a small subset of heterothallic diatoms appears to use the mt locus defined in this study?

2) The description of identification of 35 differentially expressed genes was confusing. First, Table 1 was somewhat cryptic. I think the numbers/letter combinations under the MT+ and MT- headers are library names. Also, it is not clear why all 35 genes are in the main table given that only five of these genes display mt-specific expression patterns, which is the point of the table. I would show the other 30 genes in the supplement and restrict the main Table 1 to the 5 mt specific genes.

3) Page 5, "imputable" is an odd word choice.

4) Page 5, the authors refer to the fact that MRP1 has a putative signal peptide "suggesting that the protein might be targeted to specific organelles." The authors should determine whether there is a transit peptide, which would be indicative of targeting to the plastid. Not clear what other organelles the authors are considering.

5) On page 6, the authors state "Homology searches in public databases and in the MMETSP transcriptome dataset revealed that the *P. multistriata* MT-biased genes were conserved only in diatoms (Fig. 1, A and B)." The authors should clarify whether the organisms illustrated in Fig 1B are the only species in which transcripts were detected or if the panel includes only a subset of the detected species. The distributions could be better indicated with some kind of color coding, for example, in the tree – to indicate which branches includes species in the MMETSP data set and which branches had associated transcripts. The implications of the results shown in this figure deserve more description.

6) The section on "Identification of the mating type locus" is more confusing than necessary. The authors should clearly articulate their meaning of the term MRP3 genetic locus. I believe their use of this terms refers to the coding sequence plus the ~1Kb upstream sequence. It gets confusing when they discuss alleles in terms of the upstream region in Figure 2 and polymorphisms in coding region in Figure 3 as illustrating different alleles. The authors should make it clearer how the upstream variation is linked to coding sequence polymorphisms. Also, the authors should speak to the fact that repetitive microsatellite regions (e.g., CTA) can be inherently unstable, expanding or contracting with mitotic DNA replication. The authors suggest this repetitive region may serve in a regulatory role: "growing evidence indicate that microsatellites can have a functional role acting in several processes linked to gene expression regulation." How could this work with determination of the mt of a cell, which should be binary.

7) The authors hypothesize that there are "large chromosomal rearrangements such as the one hypothesized for the N allele" that may underlie a suppressed recombination that would presumably be necessary to retain linkage of the A allele to Mt+. The authors should clarify how recombination may be repressed between the other alleles (B, M) that do not appear to be linked to chromosomal

rearrangements.

Reviewer #2:

Remarks to the Author:

This paper describes a mating type specification gene in the pennate marine diatom *Pseudo-nitzschia multistriata*. This is a potentially important finding of broad significance since it would be the first identified mating-type specification gene in a diatom and in the entire heterokont clade. The key findings were that a gene called MRP3 is only expressed in MT+ sexual strains (not MT-), that its expression is monoallelic, and that the expressed allele (designated A) correlates with MT+ differentiation, likely because the alternative alleles (M,B,N) have promoter mutations that preclude expression. A single transgenic MT- strain was described which constitutively expressed MRP3 and which showed a change of sexual differentiation to MT+, evidenced by mating behavior and differential gene expression. The core findings seem supported, but I found the presentation (figures and writing) were unclear or difficult to evaluate in several places. The manuscript needs a substantial revision and elaboration of key supporting data.

Major points

1. The authors claim to have found the mating locus in *P. multistriata*, and they have certainly found a MT-linked gene, MRP3. However, there were no data presented on linkage disequilibrium for other genes on the scaffold where MRP3 resides (sc147) which would enable the authors to show whether the MT locus is multigenic and to identify any regions of suppressed recombination. All the material needed to do these tests and identify the extent of the MT region are in the authors possession and I feel this information should be included.
2. Related to Point 1, there are genome data for other diatoms where the synteny around the MRP3 locus (if the gene is present) could have been examined to see if MT haplotype structure might be conserved.
3. One of the key claims is that MRP3 is a mating-type determining gene in *P. multistriata*. The data to support this are a single MT- transformant expressing a MRP3 A allele transgene whose mating behavior was switched to MT+. The authors did not describe how many candidate MRP3 transformants were tested, and how many showed the same mating-type reversal. Nor did they describe whether any spontaneous mating-type reversals are ever observed in culture or in control transformation experiments. Documenting that the MRP3 transgenic result is reproducible in at least one additional independent strain and that the mating type switching never occurs in control transformants is essential to support the conclusions. Furthermore, the authors need to document the mating types of their F1 progeny from the cross in Fig. 4C to show that presence of the MRP3 transgene in its new location is correlated with mating types and has become essentially a new mating locus.
4. For many readers of this study who do not work on diatom genetics there needs to be a clearer explanation of the sexual cycle, including a supplementary figure which could also serve as a visual abstract if additional information were included. A key difference between diatoms and most other well-studied microbial eukaryotes (e.g. green algae, fungi) is that mating type differentiation for diatoms is determined in the diploid phase prior to meiosis and not in the short-lived haploid gametes.

Additional points.

Figures:

5. Figure 1: In Panel A please clearly demarcate sub-clades that correspond to the representative morphologies shown on the left. Please mark the cladogram in panel A with all of the species that are listed in panel B. The white or open boxes in panel B should be described in the legend as "not detected" rather than absent. It is not possible to prove the absence of a gene, especially one that is fast evolving or that for technical reasons might be missing from an assembly or transcriptome.

6. Figure 2: Please put numerical labels under each lane in panels b-d and explicitly name the source of material in each lane and how they correspond to each other in panels b-d. Indicate what is the B label above the end lane on each gel. Indicate the inferred genotype for each relevant lane. Please mention in the main text and in the figure legend that MRP3 is an intronless gene.

7. Figure 3: Indicate the mating type and inferred MRP3 genotype in panel a. Include the inferred phasing information for the ambiguous positions in the diploid gDNA in panel a. Panel c could be laid out better. Put some columns next to the sequences to indicate the parental strain, its genotype, and the haplotype of each clone. Consolidate sequence runs from the same clone that were derived with different primers into a single row, or put them into adjacent rows.

8. Figure 4: Panel A. Don't use the Figure legend to state a conclusion. Use it to explain the presentation of data. What does the label NAT mean? What primers were used in each panel. What does H4-MRP3 mean? Panel B. Images quality should be better. What are the distinguishing features of an initial cell? Are the data quantifiable in terms of fraction of mated cells in the population? Panel C data are completely confusing. Why is there no MT+ specific band in the LV1363T3 transgenic parent? Please indicate primers, mating types of F1 progeny, and inferred inheritance of the transgene and MRP3 endogenous alleles. Number the lanes so that different F1 progeny can be described more easily. Panel D needs an internal control, preferably an intron-containing gene so that the contribution of any contaminating genomic DNA from RNA/cDNA preps can be detected. MRM2 appears to behave differently in the qRT-PCR data than in the qualitative experiment in panel D. What caused the discrepancy? Panel E. It is not clear what expression ratio means in this graph. Why not just show the expression level for each gene in the different strains normalized by the internal control?

9. Table 1: Please do a better job of annotating this table. I had to look up the mating type of each strain because the column labels for MT+ and MT- were unclear. Please label the MRM and MRP genes in this Table. Please indicate the scaffold from which each gene was derived and a reference for matching the gene ID (URL for genome hosting web site). The qRT-PCR validation for MT+ genes generally agreed with the RNA-seq data, but for MT- 4/5 genes were not validated by qRT-PCR. This raises some concern about the quality of the differential expression data and should be discussed somewhere. The heat shock factor gene(s) in the table was listed with the MT+ group but it looks like it belongs with MT- expressed genes.

10. Supplemental data: Figure S2. Include a multiple sequence alignment for the different promoter alleles. Add a Supplemental Figure that includes the MRP3 coding and UTR alignments for the different alleles. Add a Supplemental Figure with multiple sequence alignment of MRP3 predicted proteins from all species of diatoms where it has been identified by homology searches. Figure S3 resolution is very low and it is hard to read the details.

Text:

11. Abstract. A mating type locus was mapped previously in *Seminavis robusta*, so the sentence stating that this is the first diatom mating locus to be identified is inaccurate. The sentence "Only one

of the multiple allelic forms of the region upstream of MRP3 is distinctive for MT+” is unclear and needs to be rewritten.

12. Pg. 3 first sentence. Sex determination as discussed in this study is not found throughout the “tree of life”, but only in eukaryotes.

13. Pg. 3. Protists are not a taxonomic group. The references cited about sex determination in single-celled eukaryotes seems rather arbitrary.

14. Pg. 3. 2nd paragraph. This is where the genetics of sex determination need to be articulated. Note that some brown algae also have diploid phase sex determination.

15. Pg. 4. The cited reference for *Phaeodactylum tricornutum* and *Thalassiosira pseudonana* being asexual does not support this assertion. These species have the same meiotic genes as other diatoms tested. It seems more likely that sex is cryptic in these species or that the laboratory isolates have lost it.

16. Pg. 7. The data for MRP3 expression in cells above/below the SST should be shown.

17. Pg. 10. The discussion of sex determination in green algae is unnecessarily vague. The sex determining gene MID is known and can be referred to directly.

18. Pg. 10. The comparisons with brown algae are important to include, especially since some brown algae (but not *Ectocarpus*) also have diploid sex determination.

19. Pg. 10. Wording on the origin of the HMG mating type protein “from a distant lineage” is unnecessarily vague. The example cited is from an early branching group of fungi and any parallels with *Ectocarpus* would almost certainly be by convergent evolution.

20. Pg. 10. The discussion of MRP3 conservation in diatoms is imprecise. First, we do not know if MRP3 is involved in sex determination in any species outside of *P. multistrata*. Second, inability to detect the gene in more distal diatoms does not mean an ortholog is not present in these species (see point 5 above). Both of these caveats need to be included in the discussion about the evolution of diatom sex determination. Similarly, on pg. 11 the speculation about the age of the *P. multistrata* mating locus is unwarranted given the lack of available information on other diatom species.

Reviewer #1 (Remarks to the Author):

The manuscript by Russo et al. describes their discovery of a mating type locus for an important member of the marine diatoms – *Pseudo-nitzschia multistriata*. Diatoms have an unusual sexual cycle: responsiveness to an external induction trigger is linked to a reduction in cell size resulting from sequential mitotic divisions. The assumed paradigm is that cell lines will die off if sexual reproduction does not occur with a resultant cell enlargement. This is a fascinating example of the diversity of unicellular eukaryotes, with important evolutionary ramifications. The data underlying the manuscript is very solid. As described below, however, the description of some experimental results could be better clarified and the overall results placed into a larger context.

1) Heterothallic diatoms, which are the subject of this manuscript, have two distinct mating types that display Mendelian inheritance patterns. A second large group of diatoms have a homothallic life cycle where any individual cell can become either male or female gametes. The authors should place their work within this larger context. What does it mean that only a small subset of heterothallic diatoms appears to use the mt locus defined in this study?

We added a sentence in the introduction (pages 3-4) and a new supplementary figure (Supplementary Fig. 1) to provide more details about the differences between centrics and pennates. We also discuss the implications of the conservation of the *MRP3* gene more extensively on page 12.

2) The description of identification of 35 differentially expressed genes was confusing. First, Table 1 was somewhat cryptic. I think the numbers/letter combinations under the MT+ and MT- headers are library names. Also, it is not clear why all 35 genes are in the main table given that only five of these genes display mt-specific expression patterns, which is the point of the table. I would show the other 30 genes in the supplement and restrict the main Table 1 to the 5 mt specific genes.

Old Table 1 included 35 genes as this was the output of the differential expression analysis on the RNA-seq data. Only five were confirmed when we did qPCR on independent strains, as explained in the main text on page 5. The other genes that resulted regulated in the RNA-seq samples were variable when tested in qPCR on independent strains but their variable levels of expression did not correlate with the mating type.

In the new version of the manuscript, we produced a new Table 1 which now focuses on the five MT related genes, also adding information requested by Reviewer #2.

We moved old Table 1 in the Supplementary Material, it is now new Supplementary Table 1. In this we indicated strain names above the library names and used cell borders to better mark which strains were MT+ and which were MT-.

3) Page 5, “imputable” is an odd word choice.

The word has been removed.

4) Page 5, the authors refer to the fact that MRP1 has a putative signal peptide “suggesting that the protein might be targeted to specific organelles.” The authors should determine whether there is a transit peptide, which would be indicative of targeting to the plastid. Not clear what other organelles the authors are considering.

To look for signal peptides, we used two software: i) SignalP 4.0, which can predict the presence of a secretory signal peptide, a ubiquitous protein sorting signal that targets proteins for translocation across the endoplasmic reticulum (ER) membrane (Petersen et al., 2011); ii) AsaFind, a prediction tool that identifies nuclear-encoded plastid proteins in algae with secondary plastids of the red lineage, based on the presence of the conserved ‘ASAFAP’ motifs and transit peptides (Gruber et al., 2014).

The inputs for AsaFind are the protein sequence and the output from SignalP.

We added a supplementary figure (new Supplementary Figure 2) with the outputs of these analyses and the text above in the Methods, page 18.

The MRP1 protein gave a positive hit with the first software but not with the second one. This means that it has a transit peptide which allows transport in the ER but does not have the motif that in diatoms is associated with import in the plastid. Therefore, excluding import in the plastid, according to the figure below (from Gruber et al., 2017), from the ER the protein can then be targeted to the vacuole, to the Golgi or can be secreted. We predict that MRP1 is actually secreted and works as a cue from the MT+.

We have changed the text as follows:

“We analysed the protein sequence of these five genes with software for the prediction of signal peptides^{29,30} and found *MRP1* to contain a putative signal peptide that targets proteins for translocation across the endoplasmic reticulum (ER) membrane (Supplementary Figure 2), suggesting that the protein might travel through the ER and eventually might be targeted to the Golgi, to the vacuole or might be secreted³¹.”

5) On page 6, the authors state “Homology searches in public databases and in the MMETSP transcriptome dataset revealed that the *P. multistriata* MT-biased genes were conserved only in diatoms (Fig. 1, A and B).” The authors should clarify whether the organisms illustrated in Fig 1B are the only species in which transcripts were detected or if the panel includes only a subset of the detected species. The distributions could be better indicated with some kind of color coding, for example, in the tree – to indicate which branches includes species in the MMESTP data set and which branches had associated transcripts. The implications of the results shown in this figure deserve more description.

We have expanded this part in the Results (page 6) providing information that was previously compressed or hidden in the Methods.

With a blast cutoff considering only hits with an e-value $<1e10^{-3}$ and a percentage of protein sequence identity $>30\%$ (specified in Methods), for *MRP1* and *MRP3* the species in figure 1 are the only ones that were found to contain a putative orthologue in the genomes and in the MMESTP database (with the exception of a hit for *Nitzschia punctata*, now reported in the Results). For *MRM1*, which contains a HSF domain, and *MRP2* and *MRM2*, which contain LRR domains, both common domains in diatoms, we found some hits also in other diatom species. In these cases, homology was limited to the domain, which is extremely abundant, and reciprocal blasts with the protein retrieved used as query to search back in the *P. multistriata* genome found a different best hit, and/or many other hits with similar homology. In Figure 1, we only report cases in which a clear one-to-one relationship could be found. The criterion is specified in the Methods (page 18). We also provide tabular results of the tblastn for *MRP2*, *MRM1* and *MRM2* in the MMETSP database as supplementary datasets.

6) The section on “Identification of the mating type locus” is more confusing than necessary. The authors should clearly articulate their meaning of the term *MRP3* genetic locus. I believe their use of this terms refers to the coding sequence plus the ~1Kb upstream sequence. It gets confusing when they discuss alleles in terms of the upstream region in Figure 2 and polymorphisms in coding region in Figure 3 as illustrating different alleles. The authors should make it clearer how the upstream variation is linked to coding sequence polymorphisms.

To avoid confusion between mating type locus, which can actually be multigenic in many species, and *MRP3* genetic locus, that we used to indicate the transcribed region

of the gene together with its putative promoter region, we substituted the word “locus” with the word “region” when referring specifically to the gene.

To improve clarity regarding the alleles, in the new version we reconstructed the full sequence for the four alleles A, M, B and N, from the triplet repeats region to the transcribed region, this is new Supplementary Data 1 (replaces old Supplementary Figure 2 in which we had provided the sequences of the putative promoter region only). In the new Supplementary Figure 4 we now also show an alignment of the sequences of each allele, where SNPs can be visualized. This shows clearly how the upstream variation is linked to coding sequence polymorphisms.

Further information is contained in panel c in Figure 3, an improved version of old Figure 3, where we show the walking that was done amplifying and cloning overlapping portions of the locus from different strains displaying different alleles.

Also, the authors should speak to the fact that repetitive microsatellite regions (e.g., CTA) can be inherently unstable, expanding or contracting with mitotic DNA replication. The authors suggest this repetitive region may serve in a regulatory role: “growing evidence indicate that microsatellites can have a functional role acting in several processes linked to gene expression regulation.” How could this work with determination of the mt of a cell, which should be binary.

Determining the role of microsatellites in the regulation of *MRP3* expression is a very important point on which we are currently working. We expanded the discussion on pages 12-13 explaining better how a microsatellite could regulate transcription and mentioning that microsatellites can be unstable.

We hypothesize that, in cells <SST, the long stretch of CTA determines an open status of the chromatin, while the other conformations in the locus do the opposite.

7) The authors hypothesize that there are “large chromosomal rearrangements such as the one hypothesized for the N allele” that may underlie a suppressed recombination that would presumably be necessary to retain linkage of the A allele to Mt+. The authors should clarify how recombination may be repressed between the other alleles (B, M) that do not appear to be linked to chromosomal rearrangements.

Recombination can be suppressed through chromosome rearrangements but also through gradual reduction in crossover frequencies due to heterochromatinization (see Ponnikas et al., 2018). As explained on page 13, we hypothesise that the M and B alleles are linked to closed chromatin. We added a sentence in the discussion about the fact that heterochromatinization of the M and B alleles could contribute to maintaining suppressed recombination.

Heterochromatinization might be triggered by the appearance of methylation in regions where transposable elements have landed.

Reviewer #2 (Remarks to the Author):

This paper describes a mating type specification gene in the pennate marine diatom *Pseudonitzschia multistriata*. This is a potentially important finding of broad significance since it would be the first identified mating-type specification gene in a diatom and in the entire heterokont clade. The key findings were that a gene called *MRP3* is only expressed in MT+ sexual strains (not MT-), that its expression is monoallelic, and that the expressed allele (designated A) correlates with MT+ differentiation, likely because the alternative alleles (M,B,N) have promoter mutations that preclude expression. A single transgenic MT- strain was described which constitutively expressed *MRP3* and which showed a change of sexual differentiation to MT+, evidenced by mating behavior and differential gene expression. The core findings seem supported, but I found the presentation (figures and writing) were unclear or difficult to evaluate in several places. The manuscript needs a substantial revision and elaboration of key supporting data.

Major points

1. The authors claim to have found the mating locus in *P. multistrata*, and they have certainly found a MT-linked gene, *MRP3*. However, there were no data presented on linkage disequilibrium for other genes on the scaffold where *MRP3* resides (sc147) which would enable the authors to show whether the MT locus is multigenic and to identify any regions of suppressed recombination. All the material needed to do these tests and identify the extent of the MT region are in the authors possession and I feel this information should be included.

We cannot not address this point with the material included in the original version of the manuscript, since almost all the LV strains included in Supplementary Table 2 have died and the residual genomic DNA available is not enough to test a large number of markers. Whole genome sequencing of some of those strains had been attempted but since they had not been made axenic before DNA extraction the sequences obtained were mostly bacterial sequences.

We note that expression of genes upstream and downstream of *MRP3* is not differentially regulated between the MTs and neither during sexual reproduction, based on the transcriptomic data that we have on the early phases of sex (Basu et al., 2017) (in other systems it can happen that genes in the MT locus are involved in the process of mating in general). Also, the synteny analysis that we have made (see below) indicates discontinuity for the region around *MRP3* and the neighbouring gene 0020760 with respect to the flanking regions.

We have no support to make claims about suppressed recombination other than the observation that repetitive sequences tend to accumulate in sex determining regions (Ahmed et al. 2014, Sekimoto 2017, Kejnovsky et al. 2009).

2. Related to Point 1, there are genome data for other diatoms where the synteny around the MRP3 locus (if the gene is present) could have been examined to see if MT haplotype structure might be conserved.

We thank the reviewer for the suggestion.

As far as the sequenced diatom genomes, we had already reported that, using tblastn searches, MRP3 could only be found in *Fragilariopsis cylindrus*, which is very closely related to *Pseudo-nitzschia* (Figure 1 and Lim et al., 2018). In the only other available *Pseudo-nitzschia* genome, the *P. multiseriis* genome, MRP3 had not been found (Figure 1).

As to why we do not find MRP3 in *P. multiseriis* (while we tend to find the transcript in other *Pseudo-nitzschia*), one explanation could be that also in this species the gene lies in a repeat-rich region that has not been assembled properly, since the genome is still quite fragmented and of low quality due to the large portion of repetitive regions (V. Armbrust, M. Parker, personal communication, and also stats at <https://genome.jgi.doe.gov/Psemu1/Psemu1.info.html>). We commented on this in the discussion on page 12.

We performed a new analysis and added a new Supplementary Fig. 6 in which we show the syntenic analysis among *P. multistriata*, *P. multiseriis* (the only other *Pseudo-nitzschia* with a sequenced genome), and *Fragilariopsis cylindrus*. We show that the genes upstream and downstream of MRP3 conserve synteny in all the three species, while MRP3 itself and the adjacent gene 0020760 are absent in *P. multiseriis* and are present on different scaffolds in *F. cylindrus*.

We have added a sentence in the results describing these findings (page 8).

Lim, H. C., Tan, S. N., Teng, S. T., Lundholm, N., Orive, E., David, H., .. Leaw, C. P. (2018). Phylogeny and species delineation in the marine diatom *Pseudo-nitzschia* (Bacillariophyta) using *cox1*, LSU and ITS2 rRNA genes: A perspective in character evolution. *Journal of Phycology*, in press. doi: 10.1111/jpy.12620

3. One of the key claims is that MRP3 is a mating-type determining gene in *P. multistriata*. The data to support this are a single MT- transformant expressing a MRP3 A allele transgene whose mating behavior was switched to MT+. The authors did not describe how many candidate MRP3 transformants were tested, and how many showed the same mating-type reversal. Nor did they describe whether any spontaneous mating-type reversals are ever observed in culture or in control transformation experiments. Documenting that the MRP3 transgenic result is reproducible in at least one additional independent strain and that the

mating type switching never occurs in control transformants is essential to support the conclusions.

We have performed new transformation experiments and have now included in the manuscript new data for a second transformant obtained from a different MT- strain (page 9 and supplementary Fig. 9). We note that genetic transformation in *P. multistriata* is still very inefficient, and that each strain responds to selection antibiotics differently. We had to switch back from nourseothricin to zeocin selection (used in the original transformation paper Sabatino et al., 2015) because the new strain selected for transformation yielded only false positives with nourseothricin.

On page 9 we also mention that a control strain which had incorporated the resistance gene only was used in control crosses and did not show sex reversal. In general, in years of work with *P. multistriata* we have never observed spontaneous sex reversal.

We also include more data from the RNA-seq analysis that we had to show that the original LV136T3 could not be an artefact: in new Supplementary Figure 10B we show a screenshot of an IGV visualization of the LV136T3 RNA-seq reads mapped to the genome in the *MRP3* region. It is possible to note that in the transgenic strain only reads for the coding part of the gene can be found, this is because we cloned and transformed the CDS only, without 5' and 3' UTRs. In the case of a spontaneous sex reversal, the strain would have turned on the endogenous *MRP3* gene transcribing the full transcript with its 5' and 3' UTR, similarly to the wild type MT+ strains shown in the same figure for comparison. Also, the RNA-seq data can be accessed to verify that the SNPs profiles of the wild type and transformant are identical, confirming that it was the same strain and excluding any accidental contamination of the culture.

Furthermore, the authors need to document the mating types of their F1 progeny from the cross in Fig. 4C to show that presence of the *MRP3* transgene in its new location is correlated with mating types and has become essentially a new mating locus.

The F1 generation at the time of experiments shown in Fig. 4 was made of large cells above the SST that were therefore not competent for sex (we tested it). Of the original progeny shown in Fig. 4C, eight F1 strains survived until they reached the SST. With PCR we could see that three of them had inherited the transgene and five had not, when crossed with MT+ and MT- strains they all behaved as MT-. We extracted RNA and checked for *MRP3* expression which was very low irrespectively of whether or not the transgene was present. These data are now included in the manuscript (page 10) and in new Supplementary Fig. 11.

We currently do not have an explanation of why the F1 siblings with the transgene do not express it.

We note that in the original transformation paper (Sabatino et al., 2015) we had not observed frequent inheritance of the transgene. In that paper we demonstrated

inheritance of the zeocin resistance gene in the F1 with a Southern blot, however we were not able to show inheritance of other functional genes. In that study we had produced transgenic strains with an H4-GFP protein which labelled the nuclei, however when crossing these H4-GFP strains we managed to obtain only a handful of F1 cells with a fluorescent nucleus (literally not more than 10), while the vast majority of F1 cells (hundreds) did not show any signal. Similarly, we did not succeed in obtaining any F1 strains expressing GUS, another reporter gene used for those studies. We think that inheritance of the transgene is a very rare event, and that we were able to obtain relatively more zeocin-resistant F1 cells because by applying the antibiotic to the sexually reproducing culture we have selected the rare F1 cells that had inherited resistance. We have not explored this issue further, and there are no data in literature that can be used as reference since in no other diatom with transformation working it is possible to induce sexual reproduction.

4. For many readers of this study who do not work on diatom genetics there needs to be a clearer explanation of the sexual cycle, including a supplementary figure which could also serve as a visual abstract if additional information were included. A key difference between diatoms and most other well-studied microbial eukaryotes (e.g. green algae, fungi) is that mating type differentiation for diatoms is determined in the diploid phase prior to meiosis and not in the short-lived haploid gametes.

We added Supplementary Fig. 1 displaying the life cycle of centric and pennate diatoms, highlighting that gametes are the only haploid stage in the life cycle, a statement is also present in the main text on page 3.

Additional points.

Figures:

5. Figure 1: In Panel A please clearly demarcate sub-clades that correspond to the representative morphologies shown on the left. Please mark the cladogram in panel A with all of the species that are listed in panel B. The white or open boxes in panel B should be described in the legend as “not detected” rather than absent. It is not possible to prove the absence of a gene, especially one that is fast evolving or that for technical reasons might be missing from an assembly or transcriptome.

We added shades of different colors to demarcate sub-clades in panel A.

Since the tree in panel A is very dense, we added a rectangle on the clade including *Pseudo-nitzschia* and *Fragilariopsis* which corresponds to the blow up now shown in panel B.

We changed the figure legend indicating that the genes/transcripts are not detected, and also discussed this point (absence of genes) in the discussion on page 12.

Please note a change with respect to the old version: in the *P. arenysensis* MMETSP transcriptome the *MRP2* and *MRP3* transcripts had not been detected. We recently obtained a new *P. arenysensis* transcriptome in which *MRP2* and *MRP3* could in fact be found. We have added this information in panel B and in the methods (page 18).

6. Figure 2: Please put numerical labels under each lane in panels b-d and explicitly name the source of material in each lane and how they correspond to each other in panels b-d. Indicate what is the B label above the end lane on each gel. Indicate the inferred genotype for each relevant lane. Please mention in the main text and in the figure legend that *MRP3* is an intronless gene.

The figure has been modified: labels and names of samples have been added for panels b-d, B has been substituted with the word “blank”. The genotype of relevant strain has been indicated in the legend.

In the text (page 6) and in the legend it is now noted that *MRP3* is an intronless gene.

7. Figure 3: Indicate the mating type and inferred *MRP3* genotype in panel a. Include the inferred phasing information for the ambiguous positions in the diploid gDNA in panel a. Panel c could be laid out better. Put some columns next to the sequences to indicate the parental strain, its genotype, and the haplotype of each clone. Consolidate sequence runs from the same clone that were derived with different primers into a single row, or put them into adjacent rows.

The mating type, indicated by “+”, and the inferred genotype are now indicated in panel a. As far as the phasing, we verified that the only aminoacid change caused by the SNPs in the M allele with respect to the A would be a change from a Proline to a Leucine. Information about the SNPs and the aminoacid changes is not included in the main figure but it is reported in the new Supplementary Figures 4 and 7.

Panel c has been modified following the suggestions.

8. Figure 4: Panel A. Don't use the Figure legend to state a conclusion. Use it to explain the presentation of data. What does the label NAT mean? What primers were used in each panel. What does H4-*MRP3* mean?

We modified the figure legend adding the requested information and the primers used.

Panel B. Images quality should be better. What are the distinguishing features of an initial cell? Are the data quantifiable in terms of fraction of mated cells in the population?

Higher resolution images have been used for panel B. We mentioned that initial cells are the longest in the culture. We generally do not quantify the fraction of mating cells as

this is extremely variable even for the same wild type couple used over different weeks and it depends greatly on the status of the culture used. We only record presence/absence of gametes, auxospores and F1 initial cells.

Panel C data are completely confusing. Why is there no MT+ specific band in the LV136T3 transgenic parent? Please indicate primers, mating types of F1 progeny, and inferred inheritance of the transgene and MRP3 endogenous alleles. Number the lanes so that different F1 progeny can be described more easily.

LV136 is an MT- strain with alleles B/N, when transformed (LV136T3) its alleles remain the same, what changes is the random insertion of the transgene we introduce via gene gun. The amplification shown is for the polymorphic region of the MRP3 alleles, which did not change after transformation. We have added arrows similarly to Fig. 2 and the primers used to allow a better comprehension of the PCR result.

Panel D needs an internal control, preferably an intron-containing gene so that the contribution of any contaminating genomic DNA from RNA/cDNA preps can be detected. MRM2 appears to behave differently in the qRT-PCR data than in the qualitative experiment in panel D. What caused the discrepancy?

We always check our cDNAs for genomic DNA contamination amplifying a CDK-A fragment containing an intron, this control is now added in the figure. Primer sequences can be found in Adelfi et al. 2014, the reference had already been provided in the Methods (page 17).

The qPCR data show that expression of the MRM2 gene in the transgenic strain LV136T3 (green bar) is more than 5 fold lower with respect to the LV136 strain (red bar). In the RT-PCR panel, it is possible to see that the band obtained from the transgenic strain LV136T3 (lane 1) is less intense than the band obtained from the LV136 strain (lane 2), although the RT-PCR is not quantitative the two results go in the same direction. We do not see a discrepancy.

Panel E. It is not clear what expression ratio means in this graph. Why not just show the expression level for each gene in the different strains normalized by the internal control?

For qPCR data analysis we use the Relative Expression Software Tool-Multiple Condition Solver (REST-MCS), described in Adelfi et al. 2014 and in the original paper Pfaffl et al., 2002 (see Methods on page 17). With this method, one of the samples tested is set as reference condition (the zero) and expression in other samples is shown relative to this reference sample. In the qPCR shown, we had amplified four samples, the transgenic strain, the non transformed strain, an independent MT+ and an independent MT-. This last sample has been used as reference sample and it represents the 0 line. This is now explained in the legend.

Pfaffl MW, Horgan GW, Dempfle L (2002) Relative expression software tool (REST) for group-wise comparison and statistical analysis of relative expression results in real-time PCR. *Nucleic Acids Res* 30:e36

9. Table 1: Please do a better job of annotating this table. I had to look up the mating type of each strain because the column labels for MT+ and MT- were unclear. Please label the MRM and MRP genes in this Table. Please indicate the scaffold from which each gene was derived and a reference for matching the gene ID (URL for genome hosting web site). The qRT-PCR validation for MT+ genes generally agreed with the RNA-seq data, but for MT- 4/5 genes were not validated by qRT-PCR. This raises some concern about the quality of the differential expression data and should be discussed somewhere. The heat shock factor gene(s) in the table was listed with the MT+ group but it looks like it belongs with MT-expressed genes.

On the basis of this comment and following Reviewer#1's suggestions, we have kept in Table 1 only the MR genes, adding the information requested. We have moved old Table 1 in the supplementary material (new Supplementary Table 1) improving it by indicating strain names above the library names and using cell borders to better mark which strains were MT+ and which were MT-.

As far as the URL for the genes, note that in Table 1 we provide a link to the current genome browser which is password protected, the credentials are

Username: pnitzschia

Password: 30DDFA0

These credentials are given in the Methods of the genome paper, Basu et al., 2017. To improve accessibility of the genome data we are now migrating all the data on a browser hosted at the Stazione Zoologica which will not require a password, the genes are currently accessible with the following links.

NAME	ID	LINK
MRP1	24820	http://193.205.231.123/pmultistriata/?loc=comp13283_c0_seq1
MRP2	122240	http://193.205.231.123/pmultistriata/?loc=PSNMU-V1.4_AUG-EV-PASAV3_0122240
MRP3	20770	http://193.205.231.123/pmultistriata/?loc=PSNMU-V1.4_AUG-EV-PASAV3_0020770
MRM1	85380	http://193.205.231.123/pmultistriata/?loc=PSNMU-V1.4_AUG-EV-PASAV3_0085380
	41130	http://193.205.231.123/pmultistriata/?loc=PSNMU-V1.4_AUG-EV-PASAV3_0041130
MRM2	6960	http://193.205.231.123/pmultistriata/?loc=PSNMU-V1.4_AUG-EV-PASAV3_0006960

The SZN browser is being populated gradually with the tracks available in the old browser and most likely in the final version of the manuscript we will include readily accessible URLs.

Also note that in the new genome browser we were able to add the correct gene model for *MRP1* which in the old version is split in two models.

The fact that a fraction of genes is not validated by qPCR is not necessarily due to a lack of quality in RNA-seq data. Because of limited sample sizes, it happens that a real and significant difference observed with RNA-seq does not result validated in qPCR experiments conducted on independent samples of bigger sizes. This is the case for genes whose expression levels are generally variable among individuals. In an RNA-seq experiment, a generally variable gene can result significantly differentially expressed because, by chance, a group of samples in a comparison contains mainly individuals in which it is highly expressed, while the other group in the comparison contains mainly individuals characterized by a low expression for it. The qPCR validations, using independent samples, are therefore crucial to filter out these genes which are really differentially expressed between the two groups of RNA-seq samples but whose different expression levels are not associated to the tested experimental variable.

The heat shock factor was on the wrong side by mistake, this has now been corrected.

10. Supplemental data: Figure S2. Include a multiple sequence alignment for the different promoter alleles. Add a Supplemental Figure that includes the MRP3 coding and UTR alignments for the different alleles. Add a Supplemental Figure with multiple sequence alignment of MRP3 predicted proteins from all species of diatoms where it has been identified by homology searches. Figure S3 resolution is very low and it is hard to read the details.

We provide the sequences of the alleles comprising the region upstream, UTRs and CDS as new Supplementary Data 1, and present alignment of these sequences in new Supplementary Fig. 4.

We also present an alignment of MRP3 predicted proteins from all species of diatoms where it has been identified by homology searches in new Supplementary Fig. 7B.

The screenshot originally used for Fig. S3 has been substituted with an image with higher resolution generated from the browser (new Supplementary Fig. 5).

Text:

11. Abstract. A mating type locus was mapped previously in *Seminavis robusta*, so the sentence stating that this is the first diatom mating locus to be identified is inaccurate. The sentence “Only one of the multiple allelic forms of the region upstream of MRP3 is distinctive for MT+” is unclear and needs to be rewritten.

We modified the abstract taking into account these comments.

12. Pg. 3 first sentence. Sex determination as discussed in this study is not found throughout the “tree of life”, but only in eukaryotes.

We substituted “tree of life” with “different groups”.

13. Pg. 3. Protists are not a taxonomic group. The references cited about sex determination in single-celled eukaryotes seems rather arbitrary.

We rewrote the sentence referring now to unicellular eukaryotes. We added a reference to the book by Beukeboom and Perrin, 2014.

14. Pg. 3. 2nd paragraph. This is where the genetics of sex determination need to be articulated. Note that some brown algae also have diploid phase sex determination. **Here we simplified the text, we do not elaborate on any specific system as it would be impossible to be comprehensive in a short paragraph but provide relevant references. We mention more explicitly the brown algae diploid system and compare with other systems in the discussion (page 11).**

15. Pg. 4. The cited reference for *Phaeodactylum tricornutum* and *Thalassiosira pseudonana* being asexual does not support this assertion. These species have the same meiotic genes as other diatoms tested. It seems more likely that sex is cryptic in these species or that the laboratory isolates have lost it.

We have added a recent reference which supports the fact that the *T. pseudonana* laboratory strain is an obligate asexual strain while other natural strain bear signs of sexual reproduction (Koester et al., 2018). We have specified in the text that the laboratory strains are putatively asexual (page 4). In the other reference cited (Patil et al., 2015) we had indeed suggested that the presence of the meiotic kit in the genome of the two species could indicate that the species could undergo sexual reproduction in nature and that lab strains had become asexual accumulating mutations (*Spo11-2* is truncated in the *T. pseudonana* genome for instance) or that the right partner for the pennate *P. tricornutum* (pennates are generally heterothallic) had never been found (only ten isolates are used).

16. Pg. 7. The data for MRP3 expression in cells above/below the SST should be shown.

The data are now shown in Supplementary Fig. 8.

17. Pg. 10. The discussion of sex determination in green algae is unnecessarily vague. The sex determining gene MID is known and can be referred to directly.

We have rephrased the sentence and made a direct reference to the MID gene on page 11.

18. Pg. 10. The comparisons with brown algae are important to include, especially since some brown algae (but not *Ectocarpus*) also have diploid sex determination.

This has been now mentioned in the discussion on page 11.

19. Pg. 10. Wording on the origin of the HMG mating type protein “from a distant lineage” is unnecessarily vague. The example cited is from an early branching group of fungi and any parallels with *Ectocarpus* would almost certainly be by convergent evolution.

We reworded the sentence and added another reference about the frequent presence of an HMG box in sex determining genes. We had not implied any common origin of the gene between algae and fungi.

20. Pg. 10. The discussion of MRP3 conservation in diatoms is imprecise. First, we do not know if MRP3 is involved in sex determination in any species outside of *P. multistrata*. Second, inability to detect the gene in more distal diatoms does not mean an ortholog is not present in these species (see point 5 above). Both of these caveats need to be included in the discussion about the evolution of diatom sex determination. Similarly, on pg. 11 the speculation about the age of the *P. multistrata* mating locus is unwarranted given the lack of available information on other diatom species.

We agree with these observations and have therefore revised the text on page 12. We have also deleted the sentence about the age of the mating type locus as it was indeed too speculative.

Finally, we would like to highlight advantages and limitations of our system which might not be obvious for scientists working with other organisms, especially model organisms.

Compared to the other two traditional model diatoms, we have the advantage of a controllable life cycle and the possibility to obtain natural strains from a Long Term Monitoring station in the Gulf of Naples.

However, again comparing to established systems, we have a number of disadvantages.

***P. multistriata* is a delicate species, it does not easily grow in high densities and it cannot be maintained on plates, only in liquid. To perform a transformation experiment we have to grow liters of cultures. Transformation is not very efficient and proved to be unsuccessful for the majority of the strains we tested. Strain sensitivity to the antibiotics used for selection is extremely variable and we have to adjust the conditions each time that we test a new strain.**

***P. multistriata* cultures are obtained by isolating single cells from phytoplankton samples, they are not axenic. It is very hard to obtain an axenic *P. multistriata*, a culture devoided of its bacterial assemblage will die in 2-3 weeks, some strains do not stand antibiotics at all. Axenic cultures grow more slowly and it is hard to obtain biomass for DNA extraction. It is not feasible to obtain enough good-quality DNA for NGS-based strategies for the number of strains that would be needed to build a linkage map or for population genetics in a short period of time.**

Reviewers' Comments:

Reviewer #1:

Remarks to the Author:

The revised manuscript by Russo et al. is much clearer and addresses my concerns.

There are a few minor points remaining to be addressed.

- 1) There are a number of typos throughout the manuscript that should be checked.
- 2) It is too much of a generalization to state "the ancestral centrics that are mainly planktonic and the evolutionary younger pennates mostly with a benthic lifestyle." This comment is especially relevant given that the pennate diatom used for this study is planktonic.
- 3) Fig 1a – legend should state what the phylogeny is based on. 18S rDNA? It would be helpful to know where *Nitzschia punctata* falls on the tree given that this species is specifically referenced in the text.

Reviewer #2:

Remarks to the Author:

The authors have addressed the most important points raised in my previous review. I have a few additional points on the revision.

1. Not sure the authors can say they have identified the mating type locus based on the available data (Abstract line 20, Intro Line 71). They have identified a mating-type locus linked gene, MRP3, which is also a regulator of mating type. The MT locus itself could be confined to this single gene, or encompass a much larger region--up to the entire chromosome where MRP3 is located. Without linkage mapping and/or analyses of recombination and rearrangements across this region it is not accurate to say they have identified the mating locus.
2. Line 53. Spelling of anisogamous. What is the intended meaning of physiologically anisogamous? This term is typically used in reference to gamete size. Any other usage needs a clear explanation.
3. In Figure S9, the key control experiment of mating between the F4B2 parent and MT+ and MT- tester strains is missing. Also, it is not clear why the F4B2T7 transformant needed to be mated with multiple MT+ or MT- partners to establish its mating type behavior.
4. Lines 237-239. The logic of this sentence is unclear. Recent common ancestry between two species could explain why they have shared MT genes. How does this concept connect to their life histories or habitats as is implied in the sentence?
5. Line 250. Replace "transcription" with "expression". mRNA levels can be regulated by means other than transcription.
6. Lines 259/60. The accumulation of repeats or microsatellites could be cause of absent expression, but could also be a consequence of due to loss of selection for expression. Thus, their presence may not have been the driver of differential expression to control mating types.
7. Fig. 1b. Please add data for *P. multiseries* and *F. cylindrus* in the conservation matrix rather than leaving blank rows.
8. Fig. 3 legend. Please indicate mating type and genotype of strains mentioned in legend

9. Fig. 4a. Lane 3 is unlabeled in the MRP3 gel.

REVIEWERS' COMMENTS:

Reviewer #1 (Remarks to the Author):

The revised manuscript by Russo et al. is much clearer and addresses my concerns.

There are a few minor points remaining to be addressed.

1) There are a number of typos throughout the manuscript that should be checked.

We checked the text carefully to remove typos.

2) It is too much of a generalization to state “the ancestral centrals that are mainly planktonic and the evolutionary younger pennates mostly with a benthic lifestyle.” This comment is especially relevant given that the pennate diatom used for this study is planktonic.

We modified the sentence by adding that the pennate group includes also a few ecologically important planktonic genera such as *Pseudo-nitzschia* and *Fragilariopsis* (lines 53-54).

3) Fig 1a – legend should state what the phylogeny is based on. 18S rDNA? It would be helpful to know where *Nitzschia punctata* falls on the tree given that this species is specifically referenced in the text.

We added that the tree is based on 18S in the figure legend. We prefer not to indicate *Nitzschia punctata* in the figure as Fig. 1a only reports species with sequenced genomes. However, we now provide information on its approximate position in the tree in the main text, as follows: "For *MRP1*, a hit was detected also in *Nitzschia punctata* (a raphid diatom positioned in the clade above the red rectangle in Fig. 1a)" (lines 108-109).

Reviewer #2 (Remarks to the Author):

The authors have addressed the most important points raised in my previous review. I have a few additional points on the revision.

1. Not sure the authors can say they have identified the mating type locus based on the available data (Abstract line 20, Intro Line 71). They have identified a mating-type locus linked gene, *MRP3*, which is also a regulator of mating type. The MT locus itself could be confined to this single gene, or encompass a much larger region--up to the entire chromosome where *MRP3* is located. Without linkage mapping and/or analyses of recombination and rearrangements across this region it is not accurate to say they have identified the mating locus.

These observations are correct, we modified the relevant sections in the text (lines 21, 67).

2. Line 53. Spelling of anisogamous. What is the intended meaning of physiologically anisogamous? This term is typically used in reference to gamete size. Any other usage needs a clear explanation.

We have used the definition of physiological anisogamy provided by Kaczmarska, I., Poulíčková, A., Sato, S., Edlund, M. B., Idei, M., Watanabe, T. & Mann, D. G. 2013. Proposals for a terminology for diatom sexual reproduction, auxospores and resting stages. *Diatom Res.*:1-32.

‘In those diatoms in which sexually compatible gametes resemble each other morphologically but differ in their behaviour (e.g., one being motile while the other is not), reproduction is said to be physiologically anisogamous (*Neidium*, *Sellaphora Mereschkovsky*; Mann 1984, 1989b); it is recommended that such cases are always referred to as ‘physiological anisogamy’ rather than simply ‘anisogamy’, to avoid the implication that gametes differ in size or visible structure or are differentiated into sperm and eggs’

The reference has been added in the manuscript.

3. In Figure S9, the key control experiment of mating between the F4B2 parent and MT+ and MT- tester strains is missing. Also, it is not clear why the F4B2T7 transformant needed to be mated with multiple MT+ or MT- partners to establish its mating type behavior.

We changed the figure adding images of crosses with the control non-transformed F4B2 strain, and removed some of the images for F2B4T7 as it was indeed unnecessary to show many crosses with multiple MTs, a representative one is shown.

4. Lines 237-239. The logic of this sentence is unclear. Recent common ancestry between two species could explain why they have shared MT genes. How does this concept connect to their life histories or habitats as is implied in the sentence?

We agree with this observation and deleted the last part of the sentence regarding the life histories (lines 243-246).

5. Line 250. Replace “transcription” with “expression”. mRNA levels can be regulated by means other than transcription.

Done.

6. Lines 259/60. The accumulation of repeats or microsatellites could be cause of absent expression, but could also be a consequence of due to loss of selection for expression. Thus, their presence my not have been the driver of differential expression to control mating types.

We toned down the sentence which now reads as follows: “The exact role of microsatellites in the *MRP3* alleles is not clear, but one possible hypothesis is that they take part in the control of *MRP3* expression” (lines 266-267).

Future studies should test this hypothesis.

7. Fig. 1b. Please add data for *P. multiseriis* and *F. cylindrus* in the conservation matrix rather than leaving blank rows.

Fig. 1b contains information on transcriptomic data from the MMETSP project, *P. multiseriis* and *F. cylindrus* cannot be found in the MMETSP dataset and the information for them derive from genome data. We think that it would not be precise to report genome data, shown already in Fig. 1a, in the panel relative to the transcriptome data. The two species are however in the tree in 1b for reference and orientation with respect to the tree in 1a.

8. Fig. 3 legend. Please indicate mating type and genotype of strains mentioned in legend

Done.

9. Fig. 4a. Lane 3 is unlabeled in the *MRP3* gel.

Sorry for some reasons it had disappeared from the last version of the file, the new figure now contains all labels.